# Things we can do now that we could not do before: Developing and using a cross-scalar, state-wide database to support geomorphologically-informed river management

**Kirstie Fryirs**[1]*, **Fergus Hancock**[2], **Michael Healey**[2], **Simon Mould**[1,2], **Lucy Dobbs**[2], **Marcus Riches**[3], **Allan Raine**[2], **Gary Brierley**[1,4]

1 Department of Earth and Environmental Sciences, Macquarie University, North Ryde, NSW, Australia, 2 NSW Department of Planning, Industry and Environment, Water Division, NSW, Australia, 3 Coastal Systems Unit, NSW Department Primary Industries–Fisheries, NSW, Australia, 4 School of Environment, University of Auckland, Auckland, New Zealand

* kirstie.fryirs@mq.edu.au

**Data Availability Statement:** All relevant data are within the paper and its Supporting Information files.

## Abstract

A fundamental premise of river management is that practitioners understand the resource they are working with. In river management this requires that baseline information is available on the structure, function, health and trajectory of rivers. Such information provides the basis to contextualise, to plan, to be proactive, to prioritise, to set visions, to set goals and to undertake objective, pragmatic, transparent and evidence-based decision making. In this paper we present the State-wide NSW River Styles database, the largest and most comprehensive dataset of geomorphic river type, condition and recovery potential available in Australia. The database is an Open Access product covering over 216,600 km of stream length in an area of 802,000 km². The availability of the database presents unprecedented opportunities to systematically consider river management issues at local, catchment, regional and state-wide scales, and appropriately contextualise applications in relation to programs at other scales (e.g. internationally)–something that cannot be achieved independent from, or without, such a database. We present summary findings from the database and demonstrate through use of examples how the database has been used in geomorphologically-informed river management. We also provide a cautionary note on the limitations of the database and expert advice on lessons learnt during its development to aid others who are undertaking similar analyses.

## Introduction

Applied fluvial geomorphologists have produced many frameworks, approaches, manuals and recommendations to support the use of geomorphology in river management [1–18]. While understandings of river geomorphology are far from a panacea for management applications,

**Funding:** Funding for the development of the River Styles Framework over 20 years has been provided by the former Land and Water Resources Research and Development Corporation (later Land and Water Australia) Riparian Landscapes program, the Natural Heritage Trust, the Australian Research Council and Macquarie University. Applications of the River Styles Framework across NSW have over the years been funded by various NSW Departments of Land, Water and Environment, Catchment Management Authorities, Local Land Services. More recently applications have been funded by the NSW Independent Pricing and Regulatory Tribunal (IPART) to the NSW Department of Planning, Industry and Environment (DPIE).

**Competing interests:** KF and GB are co-developers of the River Styles Framework. River Styles foundation research has been supported through competitive grant schemes and university grants. Consultancy-based River Styles short courses taught by KF and GB are administered by Macquarie University. River Styles contract research is administered by Macquarie University and University of Auckland. River Styles as a trade mark expired in May 2020. FH, SM, MH, AR and MR declare no conflicts of interest.

effective outcomes are unlikely to be achieved and sustained independent from insights into geomorphic river character and behaviour, sensitivity to adjustment, measures of condition and the evolutionary trajectory and recovery potential of rivers. These are vital components of proactive planning, risk assessment and the design and implementation of management practices that work with the river, rather than against it [19–21]. Geomorphic principles document the physical template of a river system, providing a cross-scalar landscape platform upon which a host of biophysical and socio-cultural layers can be added and interpreted [22–29]. Concerns for biodiversity management, for example, build upon understandings of the dynamic physical habitat template of river systems, and associated process connections in longitudinal (upstream-downstream), lateral (channel-floodplain) and vertical (surface-subsurface) dimensions [30, 31]. Critically, a catchment perspective provides a basis to analyse the sediment regime of a river [32], underpinning assessment of system responses to disturbance events (including management actions; [33]). Building on these foundations, geomorphology provides a platform for river conservation and restoration activities, both in terms of planning and on-the-ground actions [34–36]. Despite all this promise and potential, it is not possible to make effective use of geomorphic understandings unless a reliable, coherent, systematic database is available. Systematic information bases are required to develop decision support systems that recognise diversity, consider condition and recovery prospects, and co-ordinate and prioritise actions as part of adaptive management practice [33, 37].

However, many challenges remain in the design and development of coherent scientifically-framed information databases, the process of making such databases available and Open Access, and the appropriate use of such databases in practice by suitably qualified practitioners. While a wealth of foundation understanding and knowledge may be in place, it is often fragmented and poorly coordinated. Therefore, many projects occur independently of such knowledge and understanding [38–40]. In other cases multiple approaches have been used or single approaches have been applied inconsistently and non-systematically. As work on riverscapes becomes more multi-disciplinary [25, 41] and information technology changes how data is collected, interpreted, presented, stored and made available [27, 42], it is time for geomorphologists to be become better at developing and using coherent cross-scalar databases to support geomorphologically-informed river management [43].

In New South Wales (NSW), the State Government has coordinated and facilitated the derivation of state-wide coverage of a geomorphic layer to inform river management applications. This work has been completed using the River Styles Framework [11]. NSW Department of Planning, Industry and Environment (DPIE) have now published this state-wide spatial database via Creative Commons. The NSW River Styles database is the largest and most comprehensive dataset of geomorphic river character and condition available in Australia.

The database comprises analyses of river diversity (River Styles), geomorphic river condition [44] and geomorphic recovery potential [45], thereby providing a platform for geomorphologically-informed approaches to prioritisation of management activities. In its own right, this geomorphic database does NOT provide a comprehensive solution to all river management issues that require geomorphic understanding, let alone the vast range of other information that is required. However, many management applications are likely to be compromised unless they build upon such understanding and use such databases to support the generation of integrative scientifically-informed approaches to river management [25, 46].

The NSW River Styles database presents a coherent platform for learning and adaptive management–a living database that can be readily adjusted, updated and used. On the one hand, it provides a consistent and data-rich platform for the development of geomorphologically-informed river management tools, systems and strategies, something that has been called for several decades [47]. On the other hand, it presents an opportunity to share understandings

of geomorphic river diversity, condition and recovery potential to a range of audiences who can use this evidence base to support coherent management practices. The availability of the database presents unprecedented opportunities to systematically consider management issues at local, catchment, regional and state-wide scales, and appropriately contextualise applications in relation to programs at other scales (e.g. internationally)–something that cannot be achieved independent from, or without, such a database [47].

There is no equivalent systematically-derived geomorphic database, conducted at such a scale and incorporating these layers of analysis and insight, in Australia. Importantly, this particular database extends far beyond an automated, off-site, appraisal of remote sensing imagery conducted using machine-learning models and applications [27, 48–51]. A careful mix of remotely-sensed data and generic tools has been combined with targeted field verification to compile the database. This database has largely been derived by locally-based practitioners with considerable knowledge of 'their' rivers. In many places it has been verified in the field.

Essentially, this paper outlines what the River Styles database is, how it has been derived, how it has been used, and how it can/could be used into the future to support geomorphologically-informed approaches to river management. We outline the core attributes of this dataset, summarising some of the results that emerge from an initial overview of each layer. Following this, the discussion outlines how transformations of these data can, and have been used to inform different types of river management initiatives. We highlight issues that cannot be appraised independent from such data, and the critical importance of understanding interactions between each layer of information (i.e. the use of a scaffolded information base to inform coherent management applications). The discussion provides recommendations and limitations in the use of this database, ensuring that it not perceived as a panacea, but rather as a coherent framework that can underpin systematic approaches to river management practice.

## Methods

### How the database was derived

**Development of the NSW River Styles database.** The River Styles Framework is a structured set of procedures for describing and interpreting rivers in geomorphic terms, to inform science-based river management [11]. The Framework has four stages (**Fig 1**): Stage 1 describes the geomorphic character of rivers and interprets their behaviour in terms of geomorphic process; Stage 2 determines geomorphic condition (integrity); Stage 3 evaluates the potential for a river to recover (improve in geomorphic condition) and Stage 4 identifies priorities for conservation and rehabilitation at the catchment scale [11]. The NSW Department of Planning, Industry and Environment (DPIE) added fields, including inherent fragility, to adapt the framework for its use. Using the River Styles Framework, rivers are characterised according to their present range of variability and diversity with process understandings and behavioural analysis undertaken through interpretation of form-process associations of geomorphic units [11, 52, 53] (**Fig 2**). Condition is assessed in terms of what is expected for a particular river (**Fig 3**; [44]) and recovery potential is determined according to what is realistically achievable given the character, behaviour and condition (**Fig 4**; [45, 54]).

The history of River Styles development and impact on river management practice is reported in [55]. The purpose of the River Styles Framework has always been to use foundation geomorphic principles to support scientifically-informed approaches to river management. The framework was designed to be flexible, assessing geomorphic attributes that are relevant/pertinent to a particular system. Hence, a generic, carefully scaffolded approach was designed to derive a coherent package of catchment-specific geomorphic information, relating process-based insights that can be applied to all rivers, everywhere [11, 24, 56]. Data and analyses are

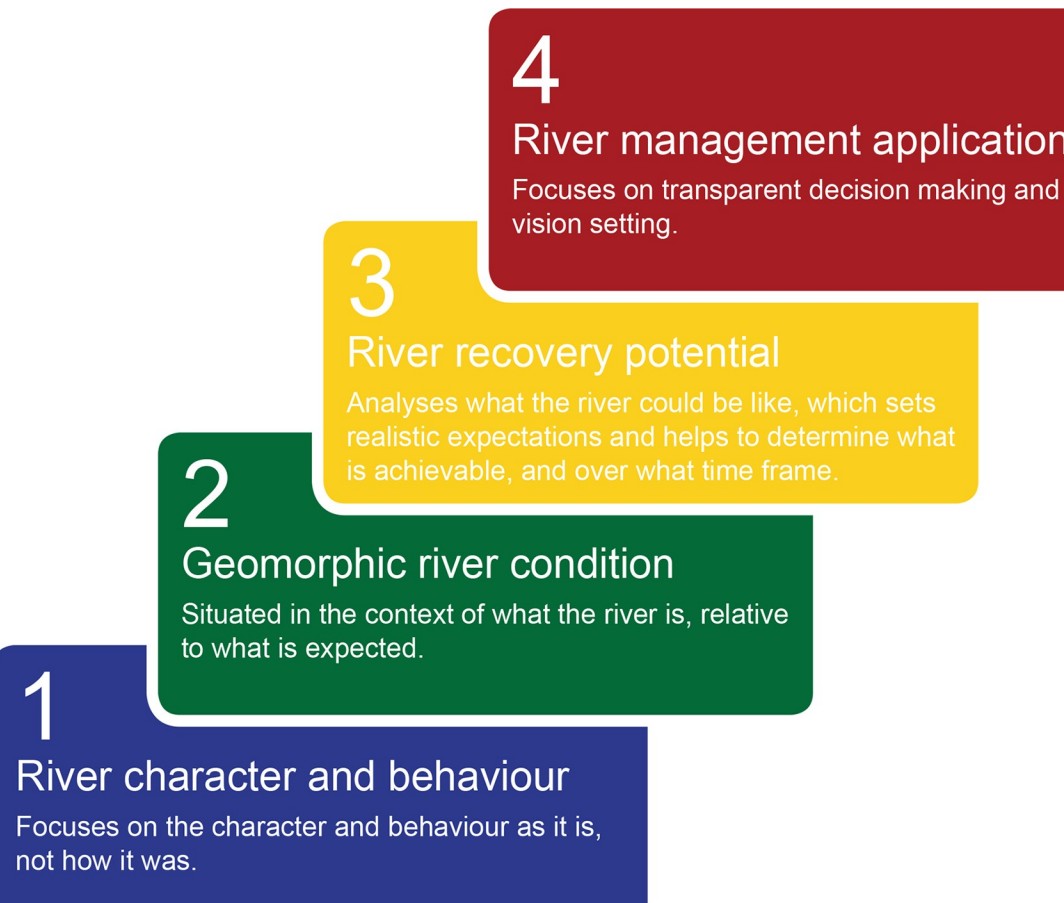

**Fig 1. The four Stages of the River Styles Framework [11].**

conducted within a hierarchical, cross-scalar framework [24, 57, 58]. Systematic approaches to data collection are used to explain spatial variability in geomorphic river character, behaviour, pattern, condition and recovery. Consistent application can help situate any given application within a much broader context, allowing consideration of the representativeness and transferability of understandings, and by extension, management decisions and responses.

The NSW River Styles database (the database hereafter) was developed by NSW Department of Planning, Industry and Environment (DPIE), in consultation with Macquarie University. It is based on River Styles reports completed by various analysts in various agencies over the past two decades (see acknowledgments). The database has spatial layers that correspond well to stages in the Framework. The river diversity layer divides rivers into reaches of the same River Style, classifying and naming them using the River Styles naming convention [59]. The condition layer classes reaches as good, moderate or poor geomorphic condition. The recovery potential layer combines assessment of both recovery potential and prioritisation to identify reaches with high, moderate or low recovery potential. These reaches are differentiated from conservation reaches, which are relatively intact (little or no recovery necessary) and strategic reaches that contain threatening processes that may impact on reaches with high conservation and rehabilitation value [45, 54]. The database also contains other attributes in its metadata that were developed by NSW DPIE Water such as fragility, change in condition, threatening processes and refugia type [60]. The database is available as an Open Access

(a) Confined gorge

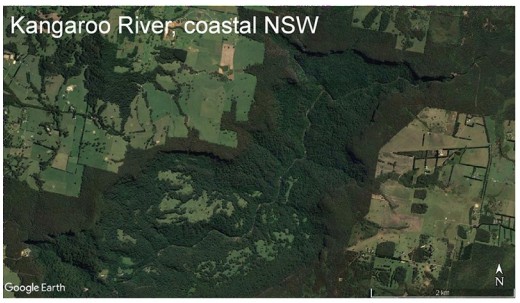

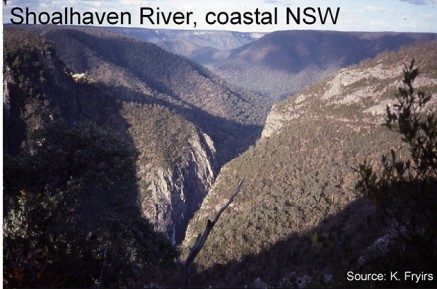

(b) Partly confined bedrock margin controlled

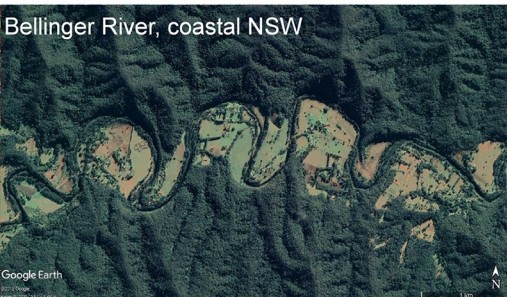

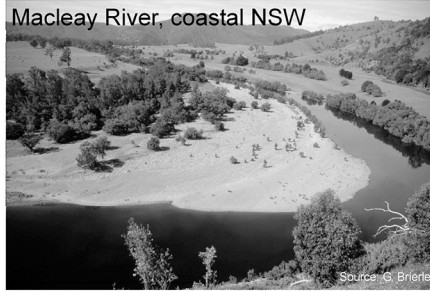

(c) Laterally unconfined continuous channel anastomosing

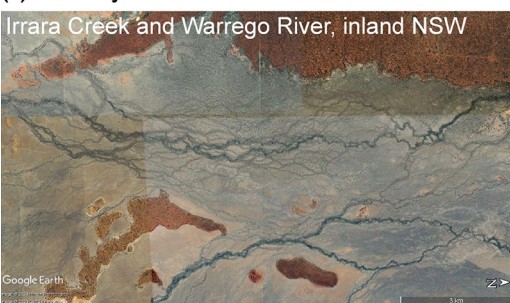

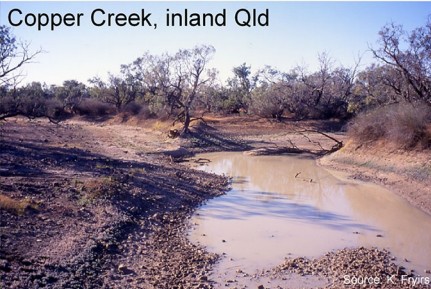

(d) Laterally unconfined discontinuous channel chain of ponds

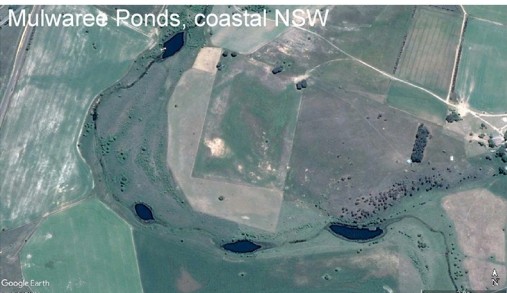

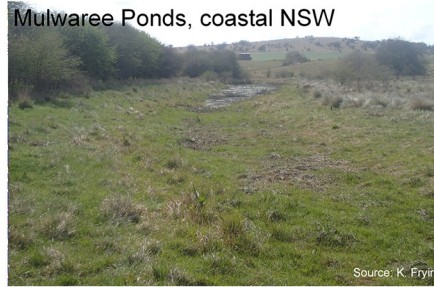

**Fig 2. Geomorphic river diversity (River Styles) in NSW.** Four of the 47 different types of rivers found in NSW. Source of photos; Google Earth, K. Fryirs and G. Brierley as noted.

product at: https://trade.maps.arcgis.com/apps/webappviewer/index.html?id=425c7364e9dc4 a71a90c4ba353b8949f.

Application trials of the River Styles Framework were initially conducted in the Bega [61, 62], Hawkesbury-Nepean and Manning catchments by Macquarie University, the NSW Department of Land and Water Conservation (now NSW Department of Planning, Infrastructure and Environment) and Catchment Management Authorities (now Local Lands Services

Good geomorphic condition

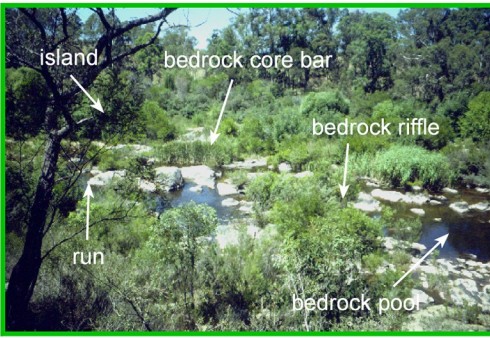

Moderate geomorphic condition

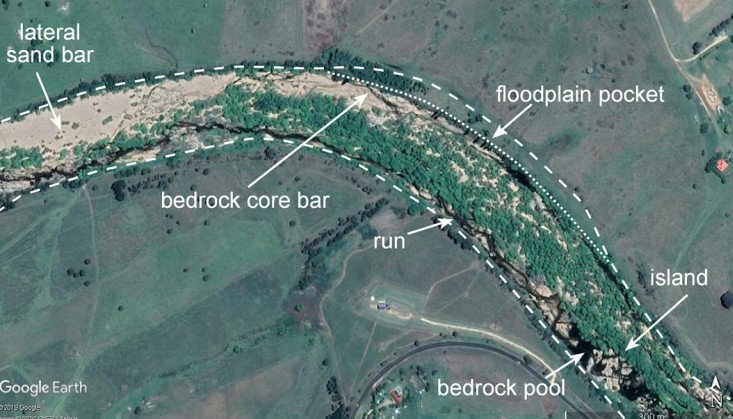

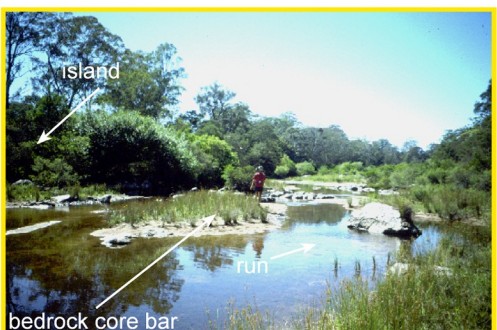

Poor geomorphic condition

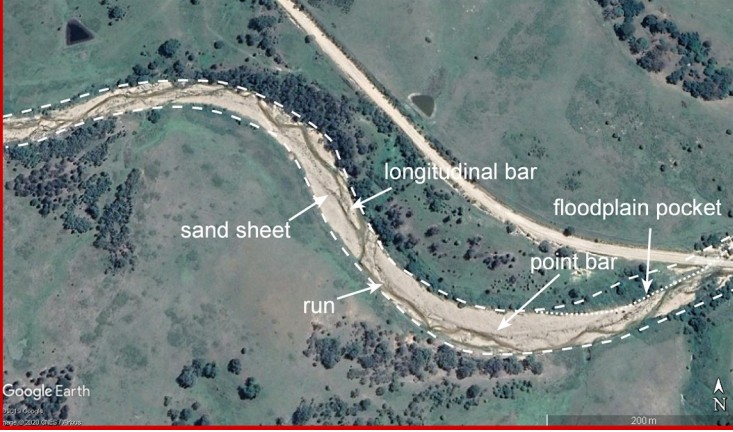

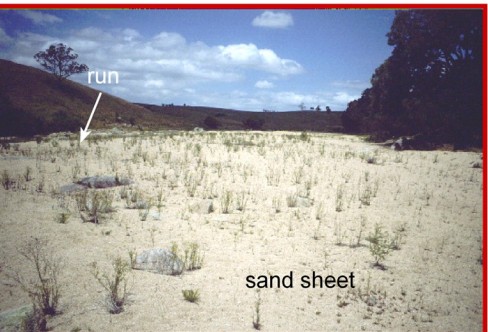

Principles for assessing geomorphic river condition
* Compare like with like
* Measure appropriate geoindicators for the River Style
* Measure condition as it is today
* Use an expected reference condition to measure against
*Identify where irreversible change has occurd by placing reaches within their evolutionary context

**Fig 3. Geomorphic condition principles for a Confined floodplain pockets sand bed River Style in Bega catchment.** Uses same examples as in Fig 4. Source of photos; Google Earth and K. Fryirs.

NSW). Subsequent roll-out across the state was conducted by various agency personnel and consultants as part of State-wide assessment and mapping between 2009–2012, funded by DPIE Water, that culminated in a major effort to complete the State-wide River Styles mapping in 2011–2012. River Styles mapping was completed in support of the State-wide

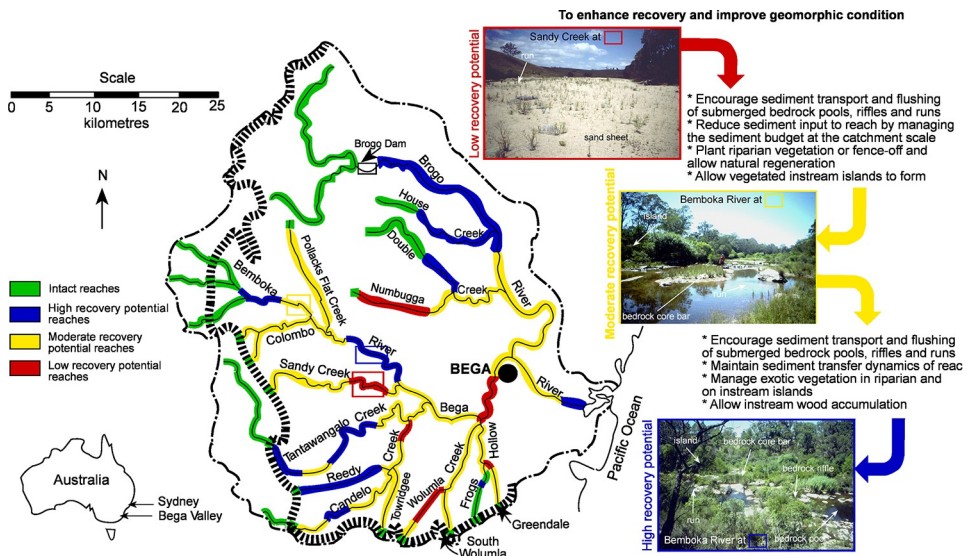

**Fig 4. Geomorphic river recovery principles and potential management actions (based on [44, 45]).** Uses same examples as in Fig 3. Source of photos K. Fryirs.

Catchment Action Plan activities the former CMAs were undertaking. An alignment process was developed whereby the State agencies brought in their products to align/inform CMA Natural Resource Management activities. Applications were mostly conducted by practitioners who had received professional development training and accreditation via the River Styles Short Course. Each report was reviewed by senior geomorphologists to ensure accuracy and consistency of application and interpretations. Applications were conducted using both field-derived and remote-sensing data, from aerial photographs in the early years, through the advent of Google Earth, to the use of Digital Elevation Models (DEMs). Importantly, 93% of the database has been verified in the field and confidence limits placed on the metadata.

## Results

### The raw data of the NSW River Styles database

The outward-facing database contains a series of maps showing findings from use of the different stages of the River Styles Framework. Because the database has a zoom-in and zoom-out function it can be used as a basis for undertaking fine-scale analyses (e.g. reach-scale) alongside state-wide assessments. Here, we simply provide a broad overview of the scale, extent and resolution of the database, and a summary of state-wide insights into river diversity, river condition, river recovery and related implications for prioritisation of management activities as part of geomorphologically-informed approaches to proactive, cost-effective river management. Embedded behind the maps is the master database that can be requested from NSW DPIE for use. To produce the results for this paper, the master data has been cleaned and processed. For example, some data pertaining to estuarine systems has been removed. **S1–S3 Tables** provide summary, cleaned data.

### The achievement: Scale, extent and resolution of the database

The database covers a total stream length of 216,600 km over an area of 802,000 km$^2$. This is an area that is 70% the size of Colombia, 65% the size of South Africa, 25% the size of India, 10% the size of Brazil, the size of the United Kingdom and France combined, the size of the

USA states of Texas and Kentucky combined or three-times the size of New Zealand. The resolution is approximately 1:1000. Reaches scale in length from 120 m to 15.5 km. It is important to note that, to date, only 3rd order streams and above have been analysed.

**River diversity.** The database identifies 47 River Styles across the state, four of which are in a confined valley setting, 15 in a partly-confined valley setting and 28 in a laterally unconfined valley setting (20 with continuous channel and eight with discontinuous channel) (**Fig 5**). This enables analyses of the frequency of occurrence (stream length) of differing types of rivers and appraisal of controls upon their patterns (**S1 Table**). Confined and partly-confined rivers dominate on and East of the Great Dividing Range whereas laterally unconfined rivers dominate West of the divide (**Fig 5C**).

The most ubiquitous River Styles by stream length were the Confined headwater (10.9%) Confined occasional floodplain pockets gravel bed (9.5%) River Styles, reflecting their position in the headwaters of catchments, where drainage density is typically greater than mid- and lower-catchment positions. The rarest River Styles by stream length were the Partly confined bedrock controlled low sinuosity gravel, Laterally unconfined multi-channel sand belt and discontinuous sand bed variants (all accounting for <0.1% of total stream length statewide). Given that River Styles analysis has only been conducted, to date, on 3rd order streams or higher, lower order watercourses (including upland valley fills) are under-represented.

There is marked variability in rivers east and west of the great divide (inland-draining and coastal catchments), and notable differences between rivers on the North and South Coast (**Fig 5**). For example, Laterally unconfined meandering fine grained rivers tend to be found on inland rivers, and coastal rivers are dominated by Confined and Partly confined rivers (some 83% of stream length in coastal catchments is comprised of confined and partly confined variants). Partly confined planform controlled meandering rivers are present in northern coastal catchments but not in southern coastal catchments, and anabranching rivers are found exclusively inland. The catchment with greatest diversity of River Styles (33) are in the Hunter, likely owing to this catchment's topographic setting and intersection between diverse geologies. Aside from Sydney Metro catchment, the Barwon-Darling had the lowest diversity with nine River Styles.

**River condition.** The database shows how geomorphic river condition varies according to attributes such as valley setting as well as allowing comparison between catchments. Across NSW, 39% of total stream length was in good geomorphic condition, 42% moderate, 19% poor and <1% with no data (**Fig 6**). Rivers in confined valley settings tended to be in the best overall condition, with the highest proportion of stream length in good condition (65%) and the smallest proportion in moderate (25%) or poor (9%) condition. This likely reflects the limited potential for rivers in confined settings to adjust, as well as their position in the catchment, often in steep areas unsuitable for intensive land use.

Partly confined rivers had the highest proportion of stream length in either moderate or poor condition (16% good, 54% moderate and 29% poor) and laterally unconfined rivers with continuous channels were in similar overall condition (24% good, 51% moderate and 25% poor). Laterally unconfined rivers with discontinuous channels were overall in better condition, with 46% of stream length in good condition, 42% moderate and 12% poor. While these results may seem surprising, these discontinuous watercourses are often remnants and 'degraded' reaches have often been channelised and therefore characterised as a different River Style (Laterally unconfined, channelised fill). Once disturbed, discontinuous watercourses often experience river change (a wholesale shift in river character, behaviour and therefore river type) [63].

Coastal catchments overall were in better condition than inland catchments, but there were notable exceptions. For example, the Hunter catchment (coastal) had the highest percentage of

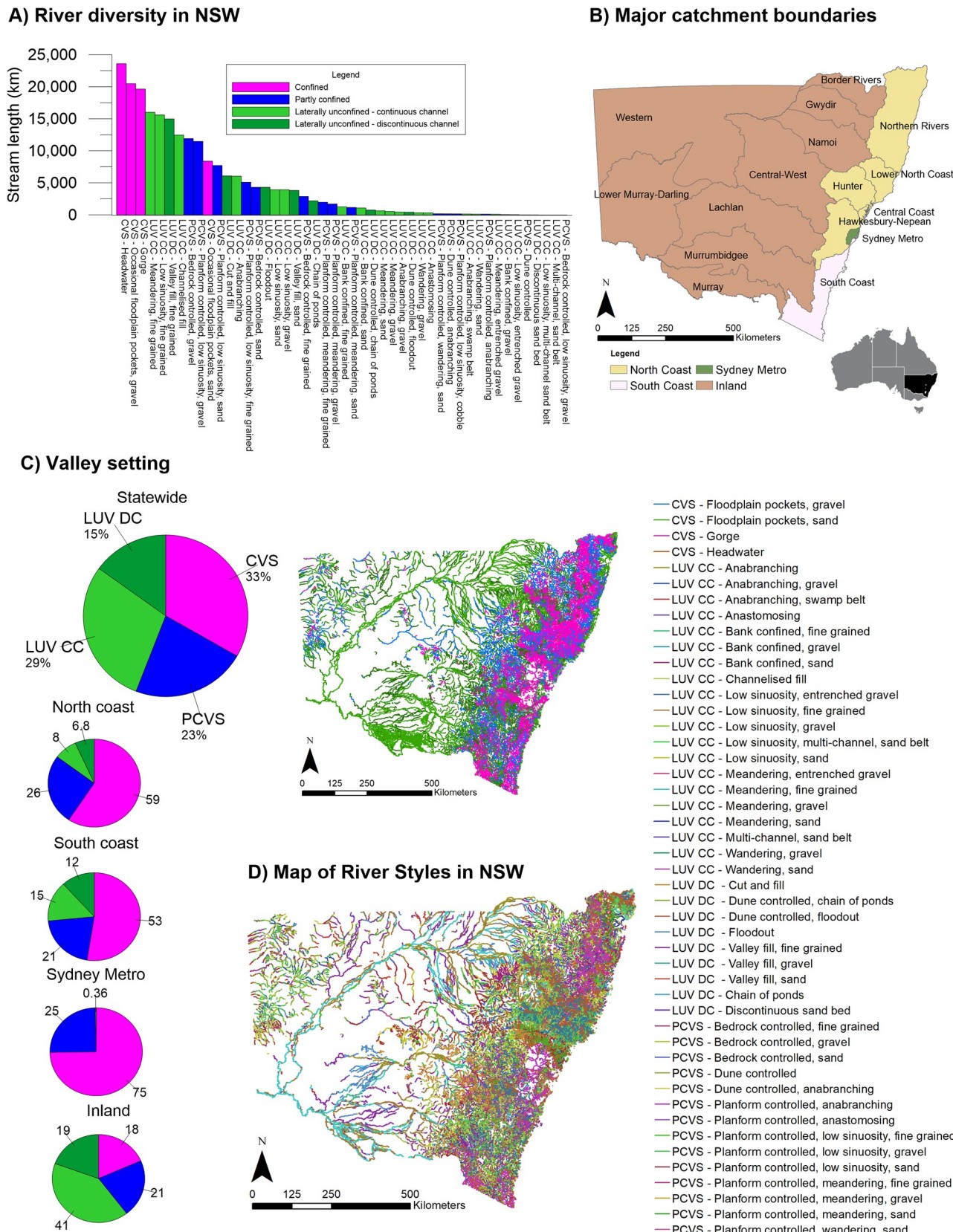

**Fig 5. The diversity of River Styles identified in the NSW River Styles Database.**

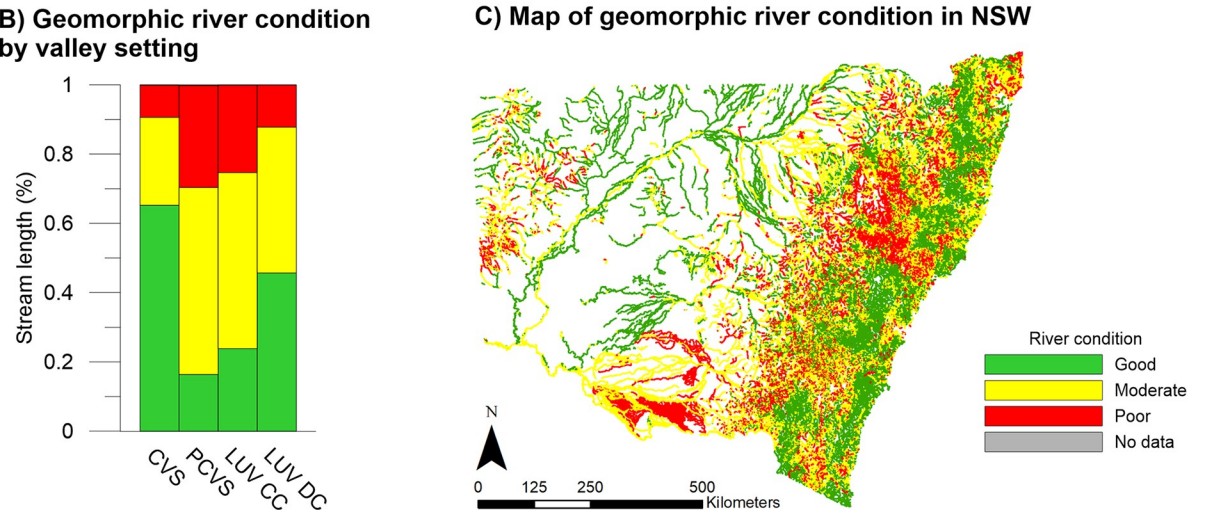

**Fig 6. The geomorphic condition of rivers identified in the NSW River Styles database.**

stream length in poor condition (45%) and the Western region (inland) had the second-highest percentage of stream length in good condition (62%). River Styles Geomorphic Condition Index (RSGCI) scores were calculated for all catchments using the method described in the NSW Government's River Condition Index [60], where:

$$RSGCI = \frac{(\%Good * 1) + (\%Moderate * 0.5) + (\%Poor * 0)}{100}$$

The average RSGCI score across the state was 0.62 out of a maximum score of 1.0 (**S2 Table**). Nine regions scored at or above average (Central Coast, Hawkesbury-Nepean, Lower North Coast, Northern Rivers, Southern Rivers, Sydney Metro, Barwon-Darling, Lachlan and Western). Eight catchments scored below average, of which six were in an inland setting. The Hunter Catchment was the lowest scoring (0.37), followed by the Namoi Catchment (0.40).

**River recovery potential and prioritisation.**   The database combines elements of Stages 3 and 4 of the River Styles Framework. This layer provides a statewide picture of geomorphic recovery potential, which, when combined with geomorphic condition, can inform prioritisation of conservation and rehabilitation efforts at various scales. Intact reaches assigned conservation status accounted for 39% of stream length (**Fig 7**). Strategic and high recovery potential accounted for 22% when combined, leaving 39% in the lower priority categories of moderate (25%) or low (14%).

Coastal catchments in general had greater recovery potential than inland catchments, with a marginally greater percentage of overall stream length with Conservation or High recovery potential and a much smaller percentage in Low condition (6.6% and 6.9% Low in North and South coastal catchments compared with 19% inland) (**S3 Table**). Catchment with significant stream length of Conservation or High recovery potential status occur in the Northern Rivers, Southern Rivers, Central West, Lachlan and Western regions. Central Coast, Sydney Metropolitan and Border Rivers also had high percentages of stream length with Conservation or High recovery potential, although these catchments have significantly less total stream length. Note that large portions of the Sydney Metropolitan catchment were not assessed due to urban development. Hotspots of Moderate and Low recovery potential included the Hunter, Central West–Castlereagh, Central West–Macquarie Bogan and Namoi catchments.

By valley setting, confined River Styles had the greatest percentage of stream length with conservation (65%) and high (14%) recovery potential as well as the smallest percentage of stream length with moderate (15%) and low (4%) recovery potential. This likely reflects the limited capacity for confined River Styles to adjust (greater resilience).

However, the second-largest percentage of stream length with Conservation status was found in the laterally unconfined–discontinuous channel setting. This group of River Styles also had the second-lowest percentage of stream length with Moderate (25%) or Low (10%) recovery potential. River Styles in partly-confined and laterally unconfined–continuous channel settings had the poorest overall recovery potential, with the majority of stream length being classified as Low or Moderate.

## Discussion

### With this database, what can we do now that we could not do before?

The NSW River Styles database provides a new opportunity for us to better 'know our rivers' and undertake geomorphologically-informed decision making in river management. Some uses of the database that are now possible, at a range of different scales are outlined in **Table 1**. Some of these uses are already occurring, while others are yet to be realised.

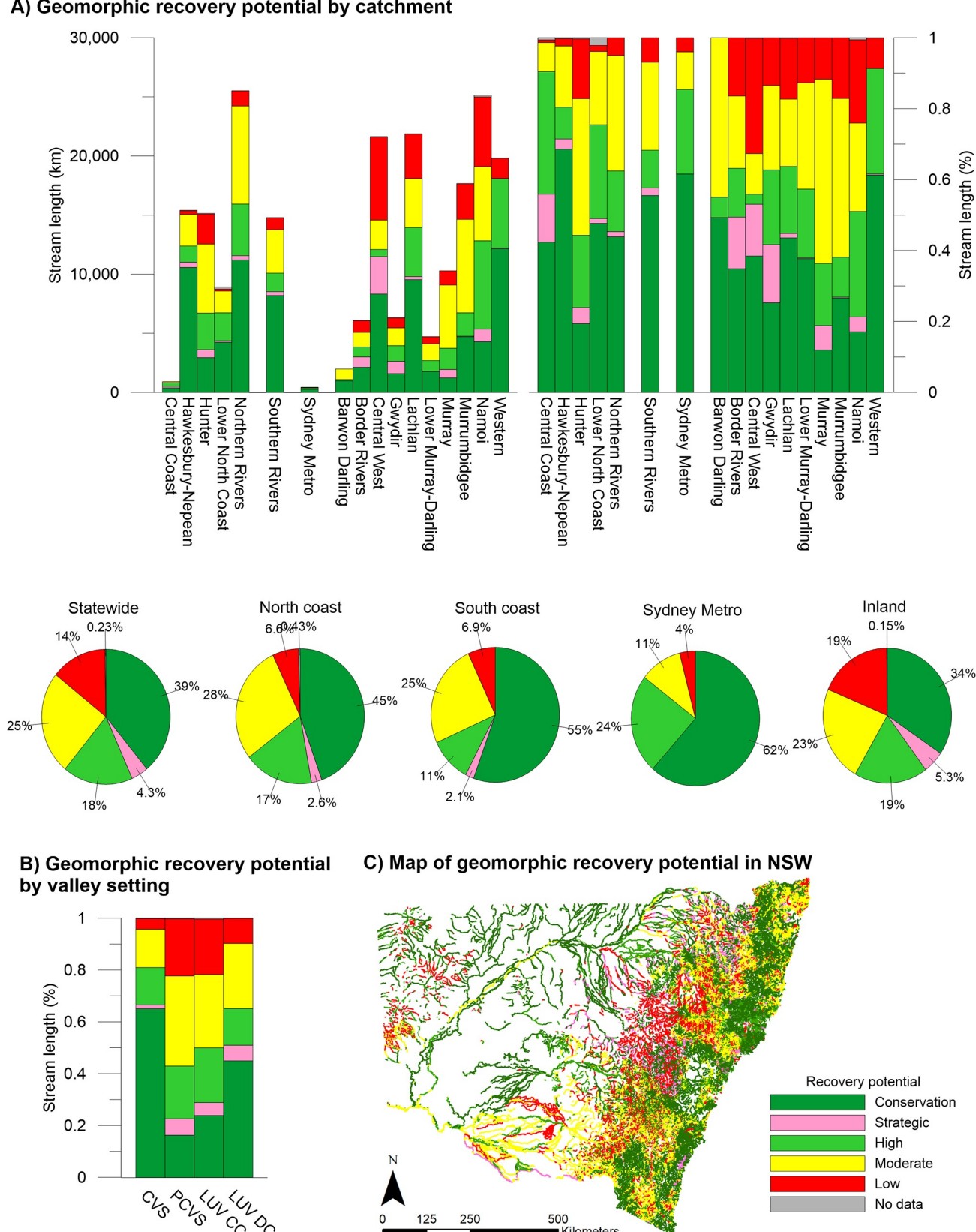

**A) Geomorphic recovery potential by catchment**

**B) Geomorphic recovery potential by valley setting**

**C) Map of geomorphic recovery potential in NSW**

**Fig 7. The recovery potential and prioritisation of rivers identified in the NSW River Styles database.**

**Table 1. Uses of database data at different scales–developed uses, potential uses and uses that are not yet realised.**

At the **site** and **reach** scale, we can. . .

- Know what type of river you are working with, and how that river behaves

- Rapidly integrate geomorphology into property management plans to make smart decisions about any rehabilitation measures that might be needed

- Understand relationships between the geomorphic structure of rivers, habitat types and ecological populations, to help protect valuable species

- Relate geomorphic understandings to local values to support asset and resource management, protection and development

At the **sub-catchment** and **catchment** scales, we can. . .

- Treat threatening processes before they become a problem

- Manage responses to disturbance events in ways that minimise both onsite and offsite impacts

- Identify and interpret the underlying causes of environmental problems, rather than just treating the symptoms

- Transfer understanding from one place to another in meaningful ways, guiding insights into when it's not appropriate to do so

- Create a vision for realistically achievable futures as part of coherent catchment action plans, designed over a particular timeframe

- Justify and verify where low- or high-cost interventions are most likely to be (in)effective and identify where small costs can make a big difference

- Integrate understandings with other datasets to identify the extent to which geomorphology is a limiting factor to river recovery (or whether other factors are at play)

- Undertake flood risk assessments and design Natural Flood Management (NFM) plans based on the geomorphic behaviour and capacity to adjust of a given pattern of rivers

At the **regional** and **state/territory** scale, we can. . .

- Integrate and align environmental decision making across agencies and disciplines via a consistent and verified baseline [e.g. 60].

- Develop management guidelines that are relevant for the types of river and prevailing boundary conditions (e.g. flow and sediment regime, riparian vegetation associations, habitat diversity and functionality, land and water management frameworks)

- Transfer understanding from one place to another (and also know when it's not appropriate to do so)

- Prioritise management activities for strategic and efficient use of resources, minimising waste in efforts to achieve the best environmental outcome and return on investment

- Undertake conservation planning (e.g. rare, threatened and endangered rivers and/or species)

- Develop and implement MER (Monitoring, Evaluation and Reporting) protocols that measure the right variables for the right river type in a consistent way

- Undertake systematic monitoring to test the effectiveness of management actions and policies

- Provide an evidence-base to inform and influence environmental and water policy

At the **national** scale, we can. . .

- Enact adaptive management

- Coordinate whole-of-government and non-government programs using a consistent information base

- Undertake state of environment audits and reporting

- Situate local, catchment and state conservation and rehabilitation goals in context of national priorities

- Undertake conservation planning (e.g. high ecological value rivers, rare and threatened species, aquatic ecosystem classifications)

- Provide an evidence-base to inform and influence environmental and water policy

At the **intercontinental** scale, we can. . .

- Make intercontinental comparisons of river type, condition and recovery

- Situate local, catchment, state and national conservation and rehabilitation goals in context of international priorities

- Fulfil international reporting, monitoring and evaluation to meet statutory obligations on the state of rivers and water resources (e.g. via UN Sustainability Goals, RAMSAR Wetlands Convention).

- Develop consortiums of databases and datasets to share understandings and support generative collaborations and networks

While the database is not perfect (no database ever is), it does provide for the first time an unprecedented big picture view of geomorphology for the purposes of river management, with diversity, geomorphic condition and recovery potential analyses contained in one place. As such, the database provides an integrating platform to do things that were not possible before. Such things include the capacity to compare like-with-like, analyse the diversity and patterns of river types across the State, understand where rivers are in good, moderate or poor geomorphic condition and where rivers have potential to recover. The database also allows users to situate their work by placing habitat-, site- and reach-scale analyses and management measures into a broader context–catchments and regions [24, 47]. This provides opportunities to scale-up management approaches and decisions to assess what works where and why. For example, it provides opportunities to combine and analyse datasets (eg aquatic habitat, threatened species and fish community mapping) and strategically target priority rehabilitation reaches with targeted funding and resources, which have to date been largely ad-hoc in their delivery [64].

The construction of a database develops a feedback process between researchers, end-users and

managers, as authorities who provide the data seek to implement the findings of the research. The process of standardisation provides a higher level mechanism for quality assurance and quality control, thus making each database more valuable. Generation and provision of the database outlined in this study can now support geomorphic and scientifically-informed, evidence-based, transparent decision making and river management, particularly in terms of prioritisation and resourcing. Each map provides foundation/baseline information that can be used for a range of purposes by a range of different stakeholders at a range of different scales [25]. Because the maps are contained within a GIS environment, they can be zoomed-in and zoomed-out to obtain information and make evidence-based decisions at a range of scales. Moving beyond the use of the database as a reporting tool or a static map, it provides information that allows users to ask a range of questions of importance for river conservation and management [65]. The ability to question across layers provides opportunities to query, interpret and explain broad patterns using multiple lines of evidence [27]. It allows users to question and interpret relationships between river character, condition and recovery potential, appraise similarity and variability, representativeness and transferability, rarity and threat. For example, visual comparison of statewide river condition and recovery potential data (**Figs 6** and **7**) suggests that rivers in poor condition were more likely to have low recovery potential in inland settings and moderate recovery potential in coastal settings. A possible explanation is provided by interpretation of the river diversity data, that considers the river type, its capacity to adjust and the materials that make up that river. It is very difficult to repair fine-grained rivers that are prevalent in inland catchments once they become degraded (**Fig 8A**). For these rivers, the trapping of fine-grained materials that support or enhance recovery is more difficult to achieve when compared to sand or gravel based rivers found in coastal catchments that can readily form benches that become stabilised by vegetation [54] (**Fig 8A and 8B**). In this simple example, information from multiple layers of the database enable deeper questioning and understanding of patterns in the data, providing one part of an evidence-base for subsequent decision making.

At the local scale (**site** and **reach** scale) the database provides a place-based information resource for local communities and landowners. People like to understand 'their' river, situating relations and understandings of 'their' river in the context of rivers elsewhere (e.g. diversity, key attributes, behaviour, condition). Promoted and used effectively, the database could provide a vehicle for community engagement and communication, supporting efforts to evaluate and look after things that matter (e.g. local values, assets, infrastructure etc).

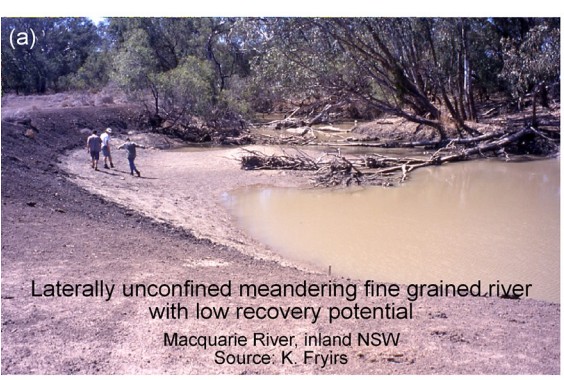
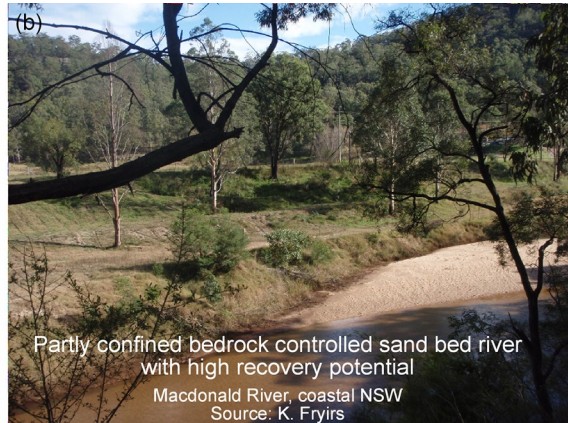

**Fig 8. Fine-grained river in inland NSW and sand bed river in coastal NSW.** Source of photos; K. Fryirs as noted.

At the local scale (**site** and **reach** scale), information in the database can be used to identify high priority and strategic reaches for conservation and/or rehabilitation (**Fig 9**). This helps in the design of rehabilitation (e.g. placement of wood and revegetation) options that are tailored to the type of river, its condition and recovery potential, ensuring that appropriate techniques are used in the right places for the character and behaviour of the river (**Fig 10**). For example, weed management and revegetation plans relate selection of species to geomorphic attributes such as the substrate, hydrological and seedbank conditions of differing geomorphic surfaces along the river [66] and wood placement is appropriate for the adjustment potential and habitat needs of the river. Using the fragility layer, processes that threaten the geomorphic integrity of reaches can be identified and decisions made about treatment responses. Working with condition and recovery shifts the emphasis from reactive to proactive river management practice, tailoring approaches that suit the conditions at any particular site or reach. Importantly, identification of reaches that are self-healing helps decision-makers to know when they can choose to opt-out of direct intervention measures, allowing the river to 'work itself out' as a form of passive management practice [54, 67, 68]. Such thinking has already been successfully applied in on-ground works in NSW conducted by Catchment Management Authorities, NSW Local Land Services, and other non-government agencies (e.g. Australian River Restoration Centre).

Working with condition and recovery at the **site** and **reach** scale also allows for identification of 'basket cases', those rivers that are in poor condition and have low recovery potential [69, 70]. Decisions can be made about whether to invest resources and finance in the rehabilitation of such reaches (or not) and if so, what is a reasonable and socially acceptable level of intervention to take [69]? Knowing whether an intervention is geomorphologically-informed or socially-driven helps decision makers and investors to set realistic expectations about the intended outcomes of management practices. As an example, the NSW database could be used to ensure such measures are applied to appropriate reaches framed in their catchment context on the one hand [33], while supporting investment in high priority and strategic reaches on the other [27].

Local scale (**site** and **reach** scale) insights from the database can be used alongside biodiversity and hydrology data to assess local habitat, flow and refugia conditions [30]. For example, the geomorphic recovery potential layer in the NSW database has been matched with hydrological stress to help set water extraction rules at the local scale [70] (**Fig 11**). Such analyses have also been up-scaled to reach, subcatchment and catchment scales to develop Water Sharing Plans [71] that are harmonised with Catchment Action Plans (CAPs) [60]. Beyond this,

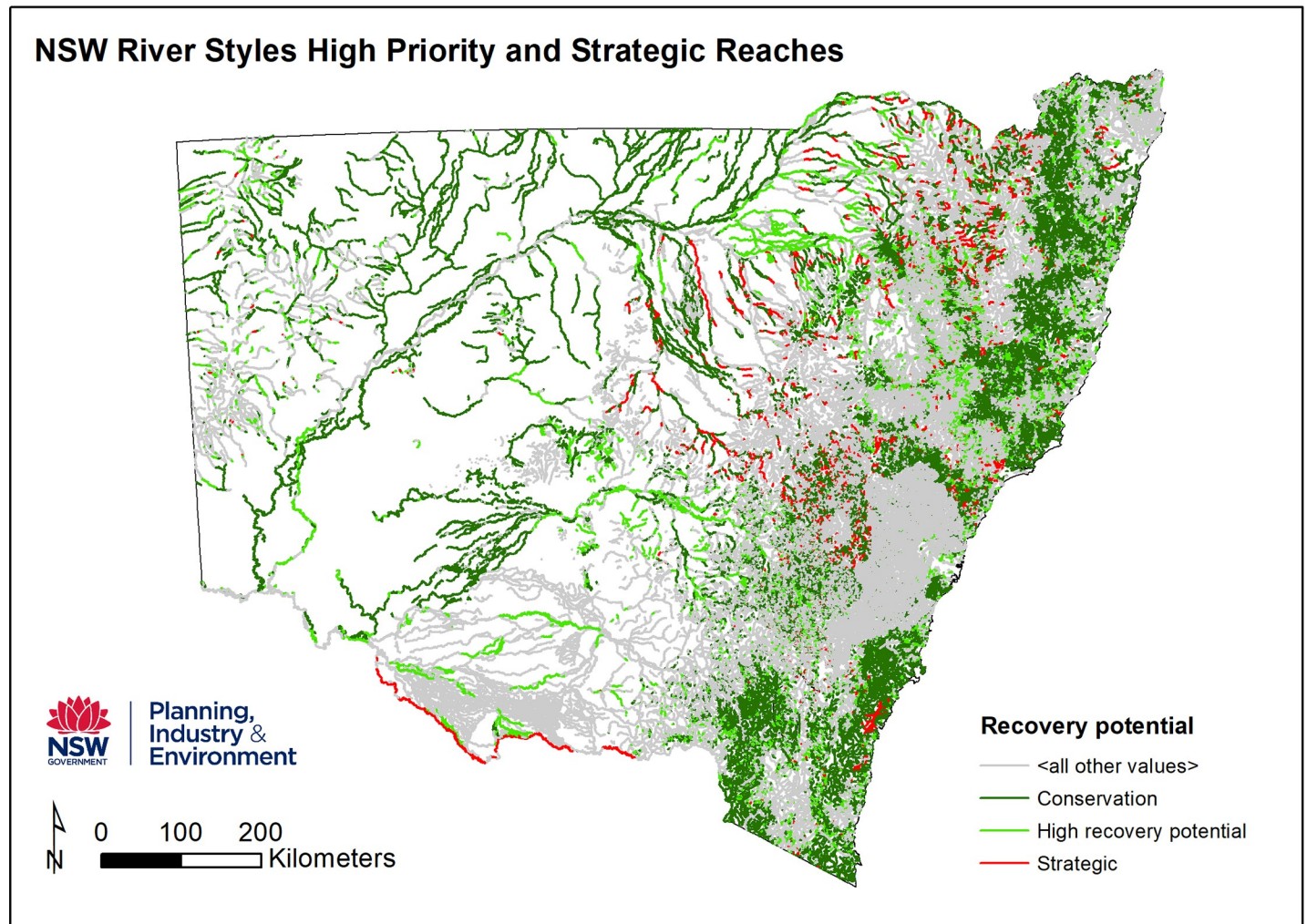

**Fig 9. High priority and strategic reaches for conservation and rehabilitation generated from the recovery layer in the NSW River Styles database.**

NSW DPI (Fisheries) has combined the database with standard statistical analysis and internationally recognised spatial distribution models to provide a delineation and spatial recognition of the status of fish communities and threatened freshwater fish species distributions across NSW [64]. The example shown in **Fig 12** uses the River Styles database to relate the distribution of fish and threatened frogs to certain types of habitats along different river types. In the USA, the Columbia Habitat Monitoring Program (CHaMP) has developed its own database for fish habitat mapping and monitoring, appraising life-cycle processes and carrying capacity of rivers from local to catchment scales [72, 73].

At a broader **sub-catchment** and **catchment** scale, the database has been used by NSW DPIE and Catchment Management Authorities to generate Water Sharing Plans (e.g. **Fig 11**), Catchment Action Plans and River Health Strategies. Such programmes inform water licensing and compliance activities, risk and threat analysis and prioritisation of on-ground rehabilitation initiatives. Incorporation of geomorphology (River Styles condition) within monitoring and evaluation programmes that apply the NSW River Condition Index (RCI) show how the physical condition of the system is a key limiting factor in river health in some instances (e.g. **Fig 13**). For example, several streams in the headwaters of the Namoi Catchment are in poor

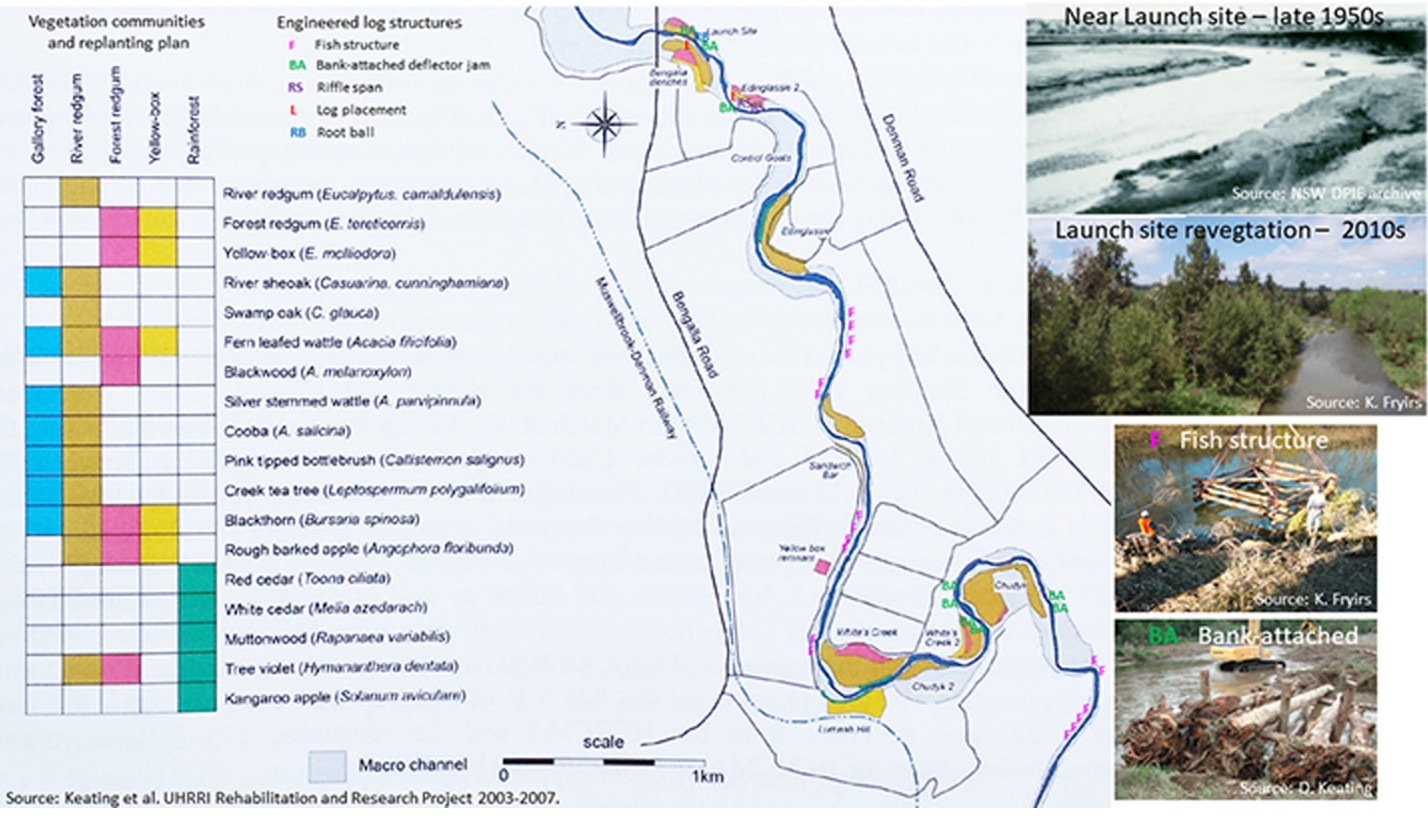

**Fig 10. River rehabilitation panning.** The Upper Hunter River Rehabilitation Initiative. Sources as noted.

overall condition. They have a very poor River Styles condition, vegetation quality is very poor, but hydrological stress is moderate. For these systems the condition of the geomorphology and riparian vegetation are limiting. To improve the overall health of these systems will require rehabilitation of the geomorphology and riparian vegetation (i.e. the physical habitat template) rather than the hydrology. Finer-scale metadata in the River Styles condition layer of the database can help users determine what to treat and what to leave alone [44, 54]. Conversely, the good to very good condition of geomorphic and riparian vegetation attributes of some western parts of the Murrumbidgee catchment are not the limiting factor to river health; rather, these streams are subjected to high hydrological stress (**Fig 13**). These examples show how interrogation of patterns in the database can aid objective and pragmatic decision making on what to prioritise and treat (or not) in river management practice. Such information can be fed directly into decision support tools to support the design and implementation of on-ground actions. Similar approaches to analysis in other parts of the world have built upon these principles (e.g. European MQI [74, 75]; UK MoRPh [76, 77]).

Applications of the database at **regional** and **state/territory** scales can help to articulate state-wide priorities and match these with regional bodies and organisations that are mandated to implement river and water management policies. This now provides a vehicle to incorporate previously overlooked geomorphic criteria (attributes of the physical habitat mosaic) within consistent, more inclusive approaches to State of Environment reporting and Regional Biodiversity assessments. Such applications now inform and influence state water policy. For example, state-wide applications of the NSW riverine High Ecological Value Aquatic Ecosystem (HEVAE) project identify and define a range of instream values and levels

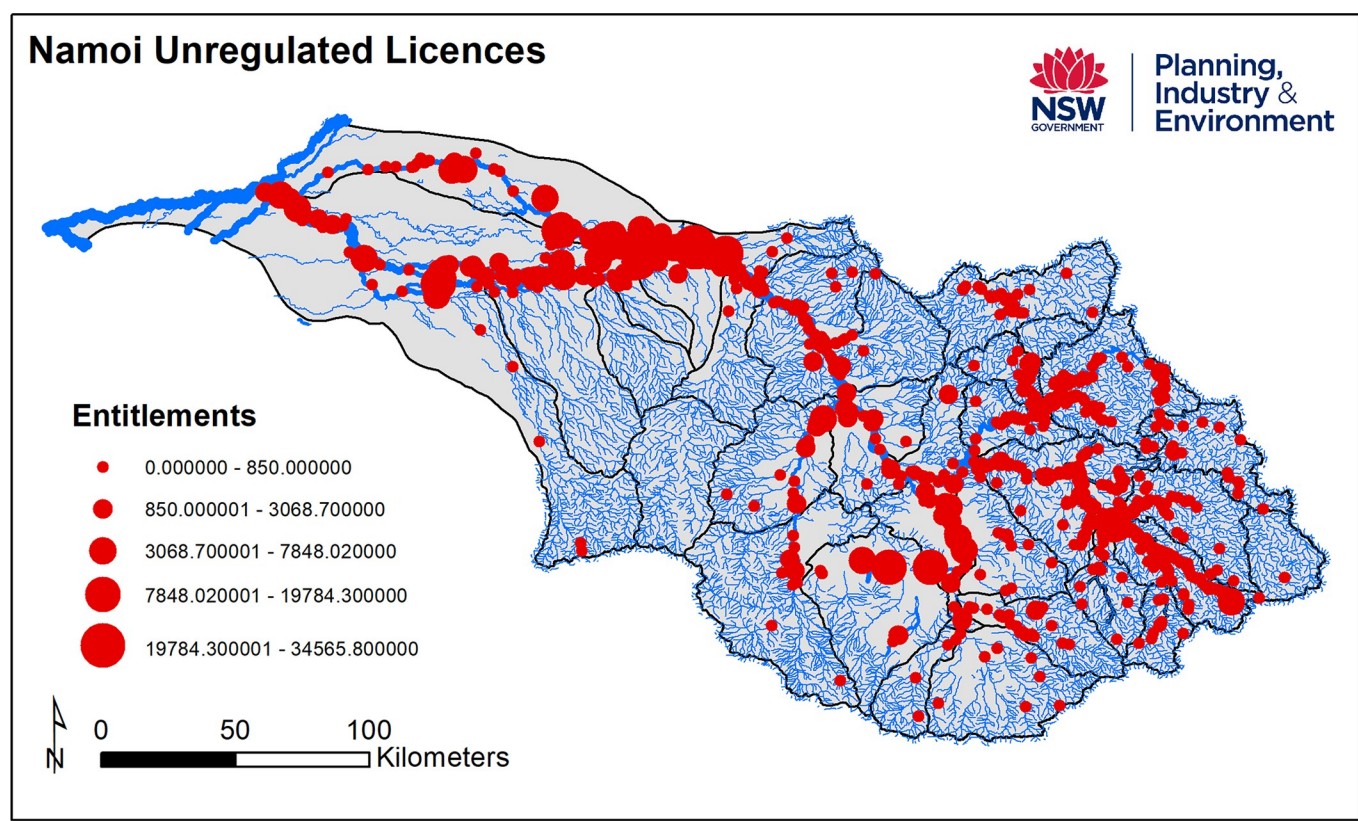

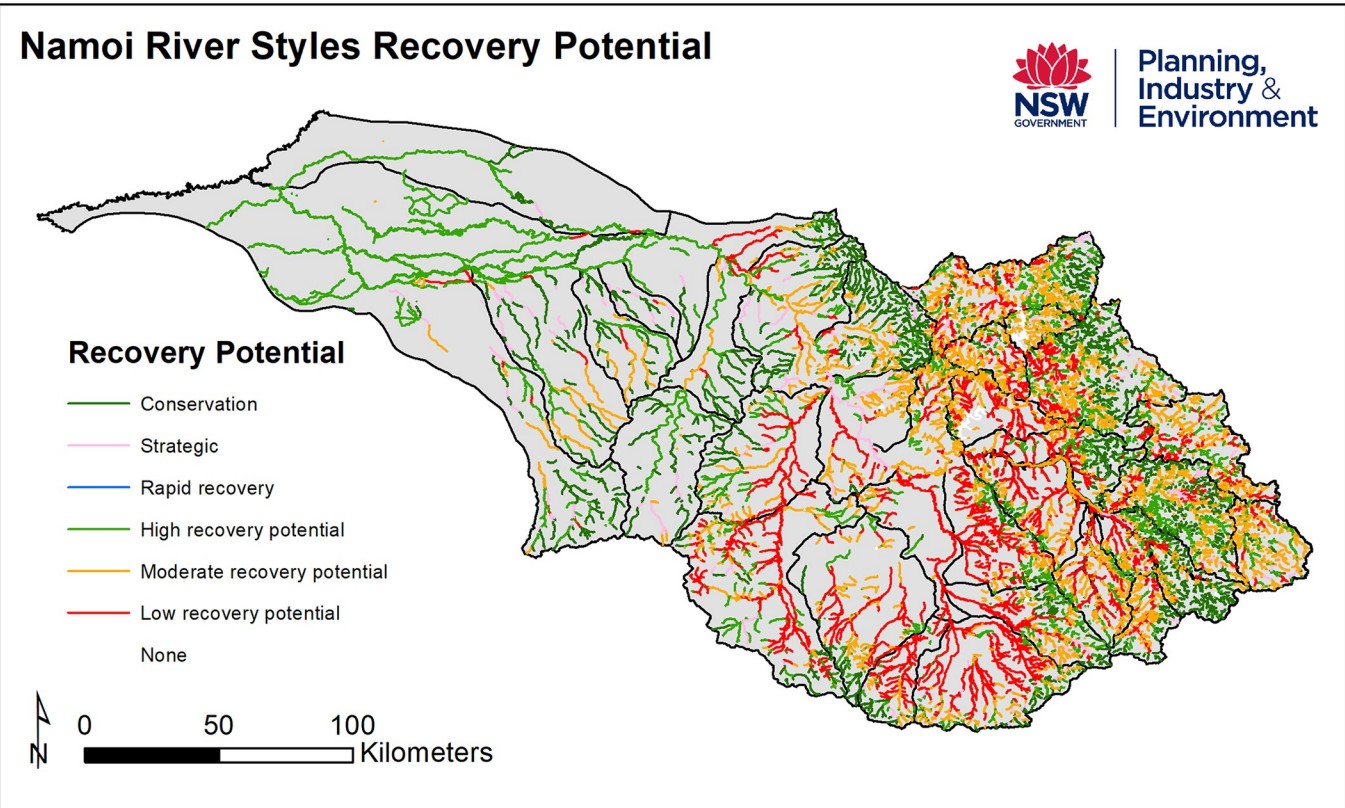

**Fig 11. Matching geomorphic recovery potential to hydrological stress for water extraction rule-setting as part of Water Sharing Plans (WSPs).** DPIE Water Risk assessment includes hydrology and geomorphic recovery potential. Shown for the Namoi Catchment in 2018.

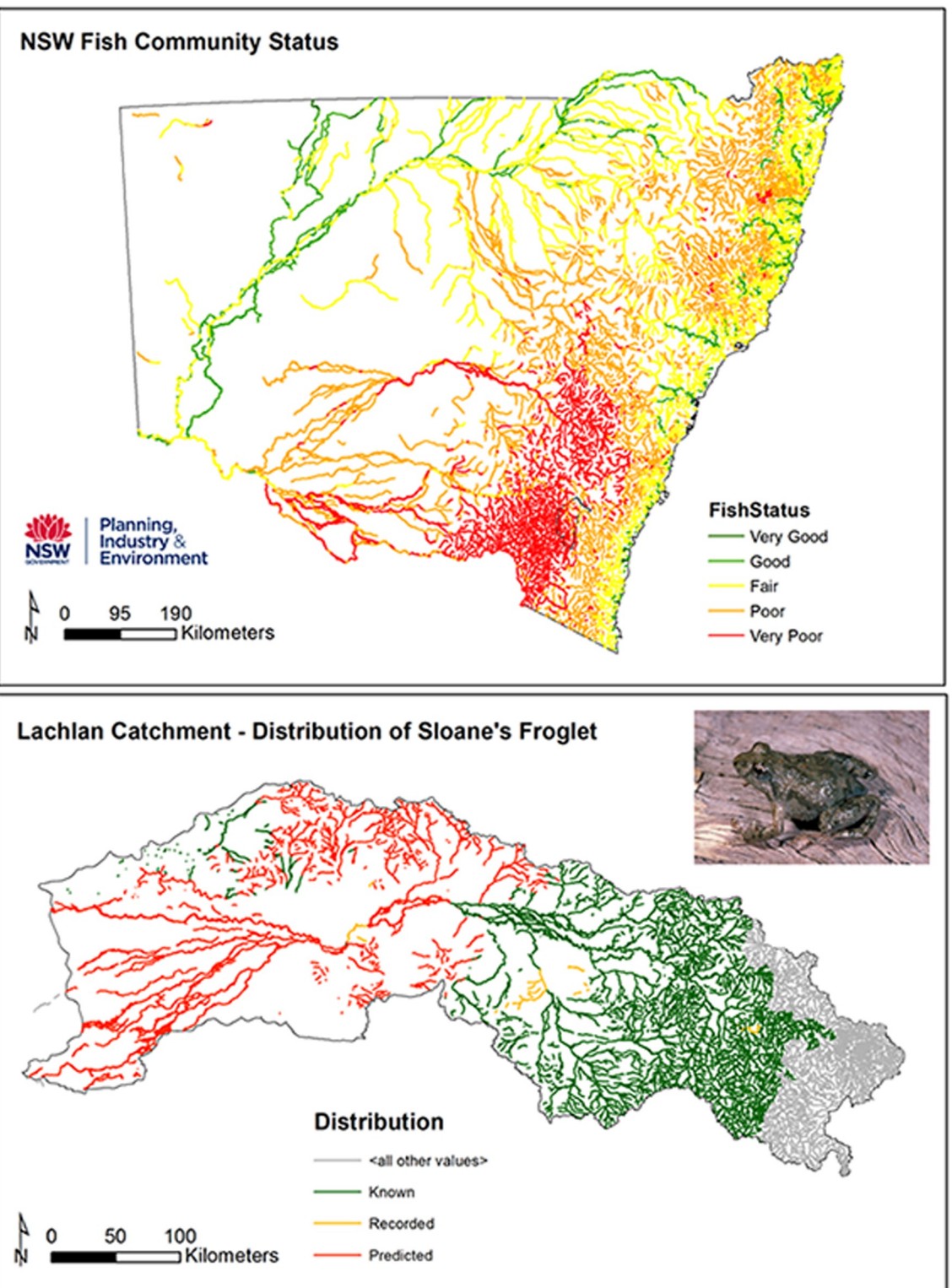

**Fig 12. Linking geomorphic river type to general fish community status derived from condition indicators of expectedness, nativeness and recruitment and indicative threatened fish distribution and threatened frog species.** Source of photograph: Sloane's Froglet from Peter Robertson at grasslands.ecolinc.vic.edu.au.

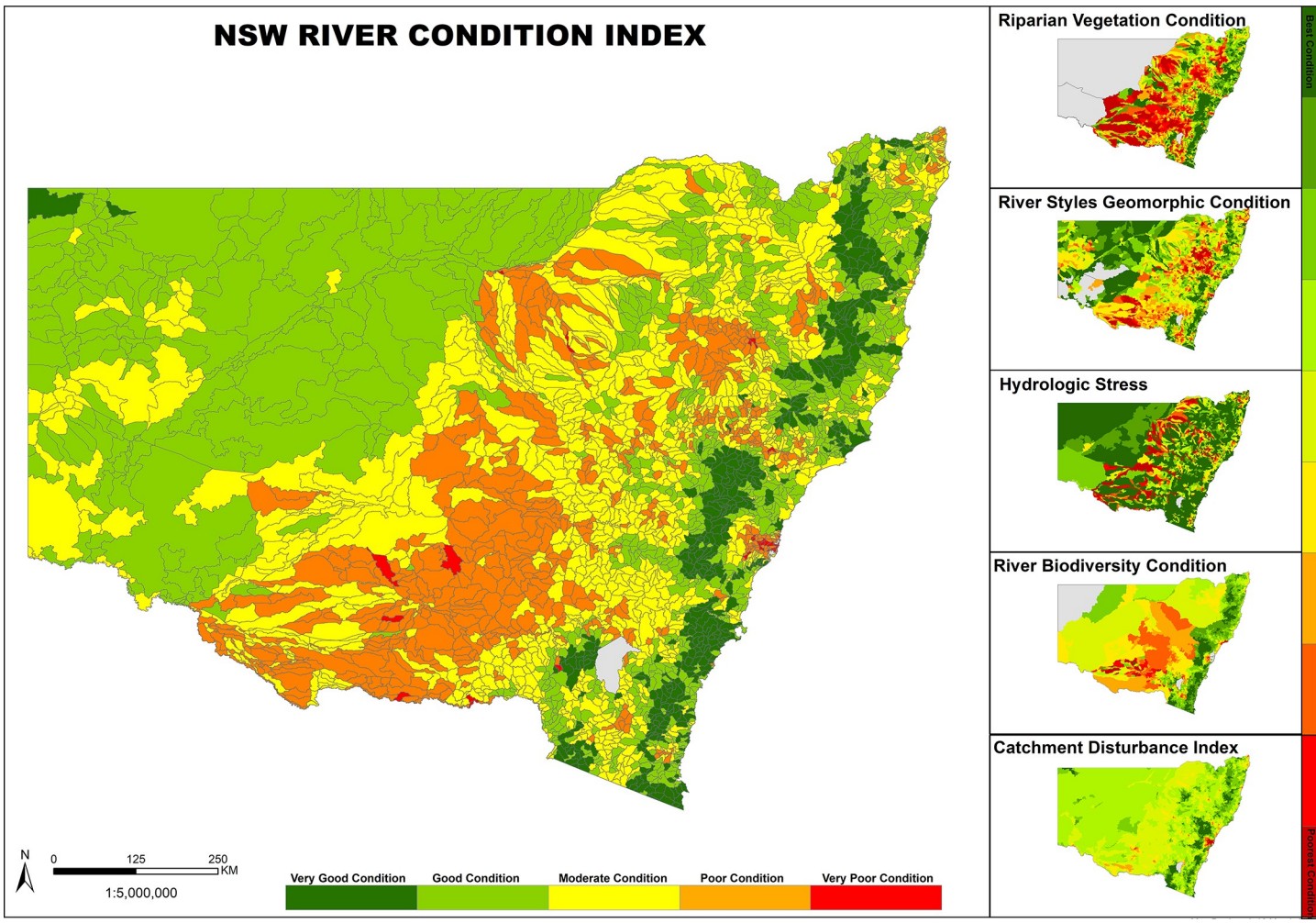

**Fig 13. NSW River Condition Index that integrates across multiple layers of attributes including River Styles (geomorphic) condition.**

of importance for freshwater river reaches in NSW (e.g. **Fig 14**). This unprecedented catchment and reach-scale resource supports prioritisation of management actions in relations to areas at risk of disturbance. Such water management practices will benefit all water users and riparian environments across the State [71]. **Fig 15** shows the level of detail that can be obtained at the sub-catchment and reach-scales for the inland draining systems, demonstrating how an integrative set of value variables can be used to undertake an assessment of risk to instream values. The example presented is for the Border Rivers-Gwydir catchment (former CMA) in the central north of the State.

Although each layer within the NSW database presents a static representation of a particular attribute, there is prospect to incorporate new and emerging datasets such as Google Earth Engine to make this a dynamic interface [78, 79]. Even in its current form, however, the database provides a snapshot in time that can support monitoring programs, helping to ensure that measures are tailored to the river type and provide a reliable relevant signal of river condition [44, 70, 80]. With a state-wide database in-hand, it is possible to address two important considerations that could not previously be appraised in a systematic manner. First, conservation targets such as distinctive, rare or threatened types of river can be managed in a proactive

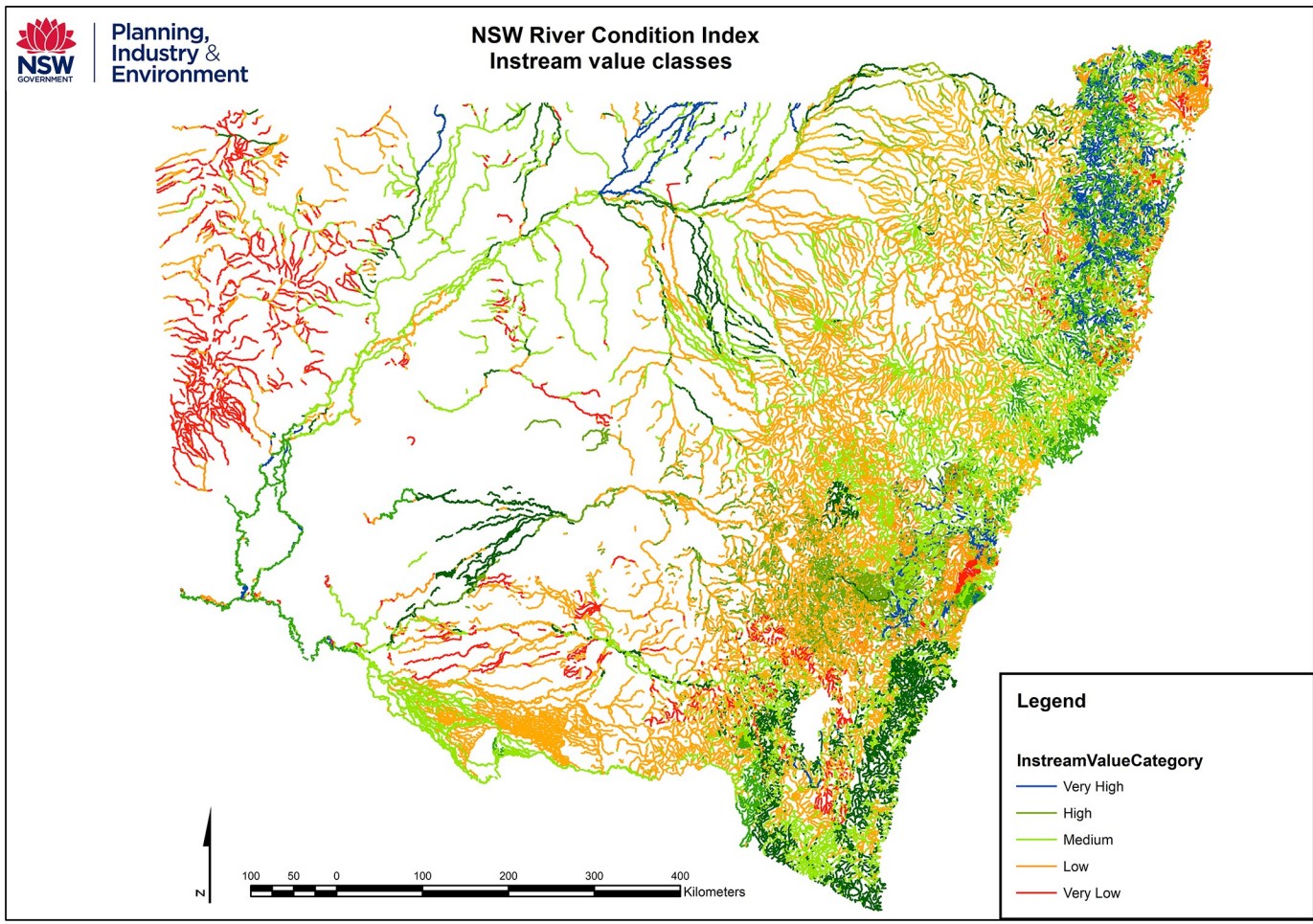

**Fig 14. State-wide High Ecological Value Aquatic Ecosystem (HEVAE) assessment that includes River Styles value as one layer.**

manner [71] (**Fig 16**). Second, incorporation of timeslices within the recovery layer can help to track changes over time (i.e. changes in colours and patterns), thereby supporting systematic appraisal of threatening processes.

At the **national** scale, the database provides a systematic reporting tool, prospectively supporting efforts to match and align state-wide priorities and reporting with National scale reporting frameworks in a more robust and consistent manner (**Table 1**). Potentially this could help situate national policies in relation to broader international or **intercontinental** programs, such as International Water programmes, conservation initiatives (e.g. Ramsar Convention on Wetlands of International Importance) or efforts to meet UN Sustainability Goals (**Table 1**). In Australia the River Styles database has been incorporated into the Sustainable River Audits and Australian National Aquatic Ecosystem (ANAE) classification (e.g. for the Murray-Darling Basin) [81] and used to align NSW DPIE Water instream value assessments with national frameworks such as HEVAE (e.g. **Figs 14** and **15**).

## Limitations and lessons learnt during the development and application of the NSW River Styles database–it's not a panacea

No dataset or database is without its limitations. Inevitably, each dataset reflects a particular set of drivers, methods and contexts that were established at a particular moment in time, then

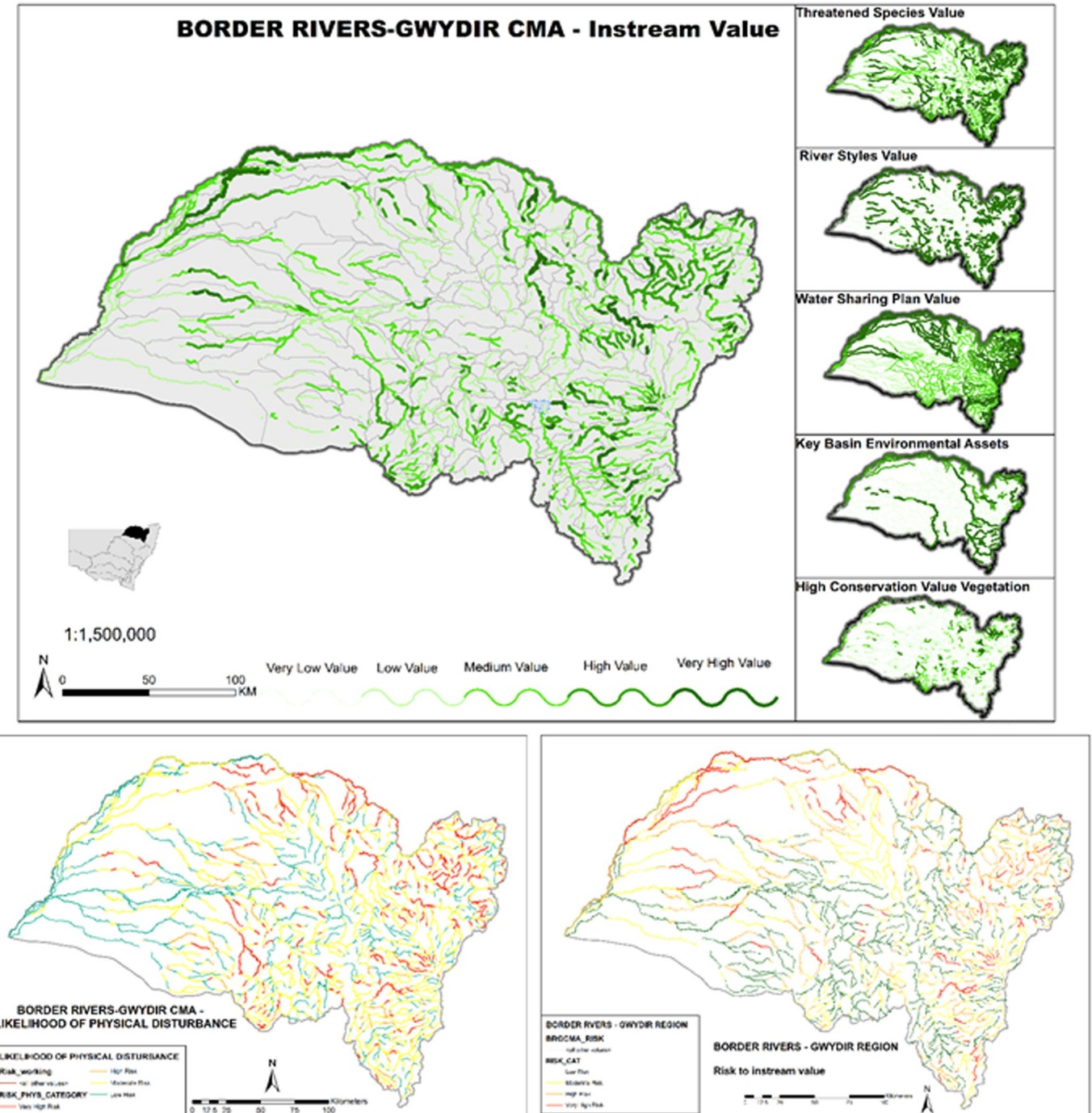

**Fig 15. High Ecological Value Aquatic Ecosystem (HEVAE) and prioritisation based on risk of physical disturbance to instream values at the catchment and reach scales.** Instream value contains analysis of the values such as geomorphic (River Styles), threatened species, water sharing, environmental assists and high conservation vegetation. The likelihood of physical disturbance to these values is then assessed to produce risk to instream values maps that can be used in prioritisation activities.

amended and updated periodically. This may reflect advances in data generation techniques or changing institutional goals, priorities, needs and legislated requirements. Here we present a brief overview of limitations in the framing, derivation and reporting of the NSW River Styles database.

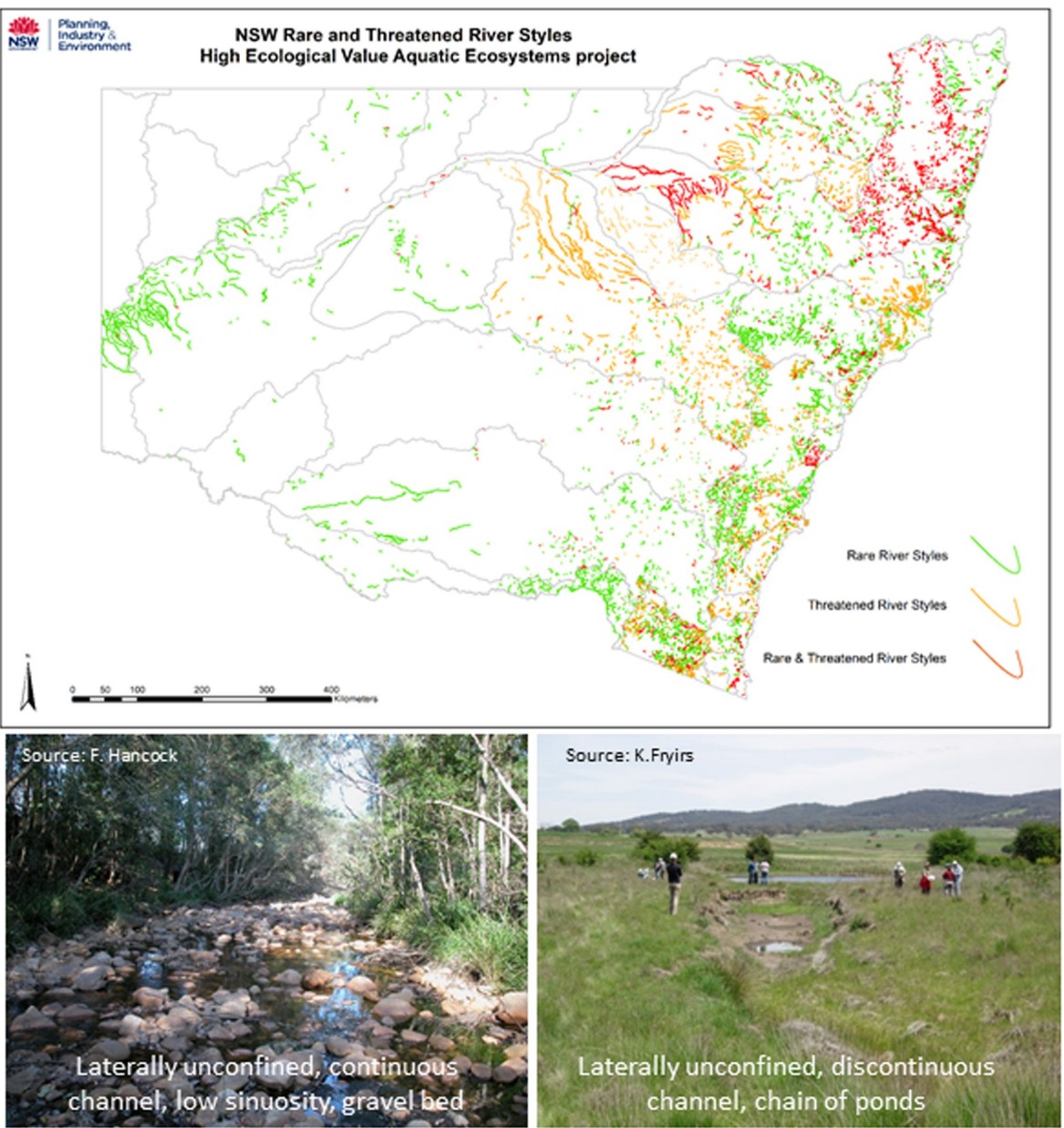

**Fig 16. Rarity and threatened River Styles in NSW.** Source of photos as noted.

While the database is coherent and directly applicable to various management issues outlined above, it is not purpose built for all applications. For example, the data layers are presented in a static form and focus largely on the structural characteristics of rivers. Much of the mapping was conducted at the scale of available aerial photographic imagery. If conducted today, maps would be produced at scales that are directly appropriate for each section of river system. In a zoom-in, zoom-out world of Google Earth, landscapes can be examined and analysed across the full range of hierarchical scales [58, 82]. Also, systematic availability of DEMs now allows analysis of all parts of landscapes [27]. Comprehensive 'whole of landscape' analyses are now possible and gaps will be filled as resources become available in future.

Analyses of process, behaviour and evolution are not presented in the database. But, it does present a coherent platform to start to undertake systematic analyses at unprecedented scales, as baseline fluvial geomorphic information is now available state-wide. For example, decadal reassessments will incorporate comparisons in geomorphic condition over time to establish trends in condition and priorities [60]. Also, it would now be relatively easy to supplement the existing database by incorporating analyses from Google Earth Engine or other remote sensing technologies to help transition the dataset into a 'dynamic entity' to inform scientific and management applications [21, 79, 83].

Given various practical considerations in the long-term development of the dataset, over a period of around 20 years, insufficient resources were available to implement effective and long-lasting quality assurance and quality control (QA/QC) procedures across the dataset as a whole (accompanying metadata are under-reported). More systematic emphasis upon assigning confidence limits to the data, and consistent application of the River Styles naming convention would also have been preferred [59]. As procedures are now in-hand to accomplish such tasks, this situation can be more readily addressed into the future, in the derivation and enhancement of this and other datasets [84]. Both the QA/QC and renaming of River Styles has been a time consuming ex-post exercise that could have been avoided if rigorous QA/QC had been developed. However, the process of 'standardisation' has provided a level of QA/QC that makes the database more useable and valuable [65]. Better metadata statements that carefully document different operators, scales of analysis and approaches to field testing and verification that were undertaken are needed [84]. Related to this, insufficient time and resources were given to verifying interpretations in the field. Time limits were placed on delivery of assessments that did not align with the spatial scale or resolution of the analyses being conducted in different places. For example, insufficient time to analyse catchments with >20,000 km of stream length or assessments conducted very quickly (e.g. in < 12 months) lead to excessive 'clumping' of River Styles or condition analyses at the sub-catchment scale.

Applications of the River Styles Framework have demonstrated that it has significant explanatory power [72, 73, 85]. However, this also comes with risks and potential for misuse, with untrained or inexperienced users using the Framework and database for purposes other than those for which they were initially derived. For example, great caution must be applied when using the database as the primary source of information for setting water availability and extraction rules, and compliance, or for justifying engineering interventions in rivers [70]. Although such applications may be perceived as required, and geomorphic information may be useful to them, they may need different forms of geomorphic insight. Just because the River Styles database may be the only available systematic, state-wide information base does NOT mean it is relevant to all instances in which geomorphic insight is required. The River Styles database is not a panacea that can be used for all riverine-related projects and to solve all problems. Hence, any extrapolations in application need to be rigorously tested to ensure that they are appropriately fit for purpose. If multi-purpose layers are to be used, they must build appropriately on strong foundations. Inappropriate applications may draw into question confidence in uptake for other purposes that align directly with initially-intended purposes. Herein lie concerns for professional credibility and integrity.

**Table 2** presents a summary of some lessons learnt in the derivation and application of the NSW River Styles database, so others who are embarking on development of their own databases can learn from our experiences. A key message is that co-ordination of activities at the science-management interface is challenging [86], especially in longer-term projects and commitments conducted in relation to inevitable (and recurrent) changes in personnel and institutional arrangements (restructuring). Long-standing research, engagement and impact collaboration was, and remains, essential for this work [55, 70, 87].

**Table 2. Lessons learned in the development and application of the NSW River Styles database.**

*Scientific issues*

• Openly acknowledge the underlying mindset and associated motivations within which the database is developed.

• Document the rationale for which the database is being generated and used.

• Use consistent terminology and naming conventions when working with an open-ended and generic framework such as River Styles–from the start.

• Develop, apply and test the use of a carefully crafted, scientifically-informed approach to collection, framing and application of data. Trialling of the approach and datasets that result in differing settings, at catchment and regional scales, can support up-scaling to develop a state-wide resource.

• Don't skip steps in the use of carefully scaffolded frameworks. In applications of the River Styles Framework, recovery cannot be assessed unless layers that analyse and document river type, evolution and condition are in place.

• Frame scientific practices and resulting datasets in ways that embrace and incorporate insights from new technologies in an increasingly data-rich digital age [88].

• Carefully consider (and document) the practitioner base and any concerns for operational bias. Sometimes River Styles analysts are splitters; others are clumpers. This is not necessarily a problem so long as scale or analysis and associated procedures are appropriately documented, such that they are reproducible. Ideally, the level of geomorphology training will also be reported in the conduct and use of such analyses.

• Check relations between information and insight derived from office-based (remotely-sensed), field-derived and modelling applications.

• Apply scientific principles of rigour, reliability and replicability, checking consistency among practitioners and reporting upon findings in a consistent, catchment-scale manner.

• Report any concerns for uncertainties and the efficacy of the data, including whether information was verified in the field. Document any assumptions and limitations in the metadata.

• Integrity and credibility are hard won, and easily lost–commitment to professionalism in practice is vital.

*Data/Data Management issues*

• Find a way to incorporate remotely sensed applications with field insights and local knowledges (a multiple knowledge lens) and reflect different levels of data and knowledge sources in data confidence and prioritisation.

• Setting up a common portal to submit and archive data and reports saves time and effort in duplication of activities and loss of data/knowledge across institutions and user groups.

• Appropriate commitment to monitoring, reappraisal and re-evaluation is required to maintain and upkeep a living database, supporting adaptive management and learning.

• Consider the use of an information management team and set of procedures to check the reliability of data and interpretations (i.e. have a code of conduct).

• Systematically roll out cases studies with appropriate QA from the start. Provide the metadata to meet QA/QC requirements, documenting who developed and derived the data, scale/conduct of analysis, over what timeframe, etc.

• Place confidence limits on the work, letting users know how the data were generated (e.g. mix of office-based, computer generated/automated and field insights; extent to which remotely sensed analysed were verified in the field). How reliable are the identifications and interpretations? Did anyone check them?

• Take care in presenting data, avoiding inherent limitations of agnotology.

• Update/amend the database as required, as new information or understanding comes to hand, so it becomes a live database–and be clear and consistent with versioning when publishing updates.

• Ensure an appropriate rationale and evidence base supports each step of decision-making processes.

*Institutional issues*

• Be conscious and aware of who is involved in processes and associated responsibilities (accountability).

• BE TRAINED before undertaking assessments. This saves a lot of pain later. Successful completion of River Styles analysis cannot be completed from a book or taught via email. It pays to invest in professional development or use a qualified geomorphologist.

• Carefully constructed approaches to professional development are required to accompany use of the database and the changing skillsets and approaches needed to analyse, interpret and use such data.

• Prioritise commitment and resources to develop and maintain a spatial analysis team, with associated infrastructure (personnel, hardware, software).

• Create and document procedures for selection and training of personnel.

*(Continued)*

**Table 2.** (Continued)

| |
|---|
| • Maintain recurrent contact between researchers and management agencies through co-ordinated approaches to organisational arrangements, with clear articulation of responsibilities and expectations. |
| • Maintain networks between government, industry and community groups to improve information flow, maintain community engagement and monitoring and evaluation of river management. |
| • Carefully consider cross-institutional support to develop a whole-of-government approach (this is not an add-on, this is integral to the development and conduct of the work). |
| • The scale and reliability of data and interpretations must be checked before considering any applications in relation to compliance or consents-based issues. |
| • If prioritisation or strategies are amended (not heeded), provide the reason(s) and reasoning. |
| *Co-ordination of applications* |
| • Ensure applications are fit-for-purpose. |
| • Put procedures in place to co-ordinate applications and extensions, outlining what the tool has been designed for and how it is intended to be used. |
| • The River Styles Framework was established to respect the inherent diversity of rivers, not make rivers the same. Care should be taken in application of engineering structures that suppress river behaviour. |
| • Take care in relating datasets to insights derived from other perspectives or disciplines. |
| • Take care with representativeness and transferability of the data generated (knowledge mobilities and mutability). |
| • Careful reporting of activities is a professional responsibility, not an optional add-on. |
| *Some practical realities* |
| • Patience, persistence and diligence are required in setting up collaborations, maintaining relationships, designing and activating approaches to dataset development, and implementing and updating the resource base. This should be recognised in workloads. This work cannot be done in-kind. |
| • It takes time to develop such databases. This often entails complex deliberations and negotiations. Investment is worth it in the medium to long term. |
| • Recognise that while efficiencies can be made in places, this should not occur at the expense of coherency and scaffolding of information. |
| • Be discursive but avoid farnarkling. Pay appropriate attention to detail, without getting hung up on irrelevant distractions. Note: Farnarkling is an Australian slang term meaning "a group activity where everyone sits around discussing the need to do something, but nothing actually happens". |
| • Be flexible/adaptable, open to new information sources and approaches. |
| • Be honest. If something isn't working, change it. |

## Moving forward: Situating the River Styles database for use in science and management

It is one thing to have the NSW River Styles database, but quite another to contemplate the usefulness and practical application of this valuable resource. With a decent information base in-hand, the potential for geomorphologically-informed river management is high. Prospectively, the NSW River Styles database can enhance integrative river management in ways that are not possible with unconsolidated or fragmented data [46]. Untold benefits, impacts and end-users, many unimagined at this time, could emerge in the future [55]. However, as we have learnt on this journey–training is critical. Knowing what such databases hold and how to use this information 'correctly' is critical. Suitably qualified and trained professionals are needed if geomorphologically-informed river management is to achieve best practice.

River management is much more than a collection of projects; it is an ongoing commitment, obligation and responsibility [38, 39, 88]. Appreciation of the proper use and value of the River Styles database requires ongoing engagement and education. Improving community comprehension of risk to river integrity and consequences of river degradation needs ongoing contact and education [89]. Ongoing commitment by government and industry is required to

ensure that the River Styles database and the assessments that underpin it remain a living project, meeting practitioner, industry and community needs and expectations [90].

Perhaps one test of such a database lies in its capacity to act as a medium for relating local knowledges to geomorphologically-informed river management practices. It can act as a pivot for developing a collective sense of riverscapes, that relates place-based local connections and knowledges to broader scale (state, national or international) programmes and applications [91–94] and support participatory engagement and collective commitment and championship of a duty of care in river management [86, 95–98].

The construction of the database has, in itself, developed a feedback process between researchers, practitioners, managers and end-users [65]. Effective, coherent and inclusive approaches to river management now need to build upon, and value-add to, this collectively-owned information and the understandings that come with it. As the NSW database is Open Access, there is considerable opportunity to develop a coordinated whole-of-government (and non-government programmes) that use a consistent information base [97]. In turn, the database could provide a sense of common ground that helps facilitate cross-scalar, cross-institutional approaches to river management, acting as a communication platform between people, within communities and across organisations–domestically and internationally.

Proactive and strategic measures are not possible in the absence of such databases and the understandings that accompany them, inhibiting prospects to determine what is realistically achievable through management efforts, and what measures must be applied to achieve and sustain the outcomes pursued [32, 98]. Without such databases, it is not possible to 'be ready' with capacity to 'hold steady at times of crisis', limiting management responses to renewed reapplication of reactive measures, many of which we know will not work in the long term [88]. Reactive management is costly and restricts future prospects. Even more alarmingly, such reactive measures set path dependencies that compromise prospects for more generative programmes into the future.

While the NSW database has been developed using a combination of 'traditional' desk-top and field methods and is currently presented as a set of static maps, its value should not be diminished. Use of the River Styles Framework has contributed to the adoption of process-based river management in NSW [c.f. 36, 99]. To produce the outputs presented in this paper requires interpretation of river behaviour, evolution, patterns and (dis)connectivity that extend well beyond the production of a map [100–102]. Nowadays however, new technologies and analysis techniques simply make the job of developing some parts of a River Styles database more time and cost-efficient (e.g. using Google Earth and Google Earth Engine (GEE) along with Geomorphic Change Detection (GCD) and automated tools for geomorphic analysis of rivers) [78, 88, 83]. We are not yet at a situation where databases such as the NSW database can be fully automated, but certain parts of the analysis can be semi-automated using available and emerging 'plug-in' tools for analysis [88] and by tapping into large-scale remote sensing datasets and toolboxes held in Open Access or consortium-based repositories [27, 51, 83]. Such advances provide a fantastic opportunity to now juxtapose a static map with analyses conducted using GEE or GCD (for example) to display river adjustment, behaviour and change over time [21, 79]. Such analyses are sparking a wave of innovation and collaboration in the development of semi-automated tools to support geomorphic analyses of rivers, allowing new science questions to be asked about river diversity, behaviour, condition, trajectory and forecasting. However, it is rare that approaches to generation and use of geomorphic datasets are undertaken using a coherent framework that is applied consistently and at the scales needed for catchment, state, national or international planning and decision making. The NSW database provides an example of what can be achieved and what is needed moving

forward. It provides a fantastic basis for considerable value-adding, extension and multi-use uptake by government, industry and community in years to come.

## Conclusion

The NSW Government has facilitated state-wide delivery of a geomorphic layer for the analysis of rivers. This database has been crafted and implemented using procedures from the River Styles Framework. Although this does not provide a comprehensive answer or solution to all river management problems, we are unaware of an equivalent database produced to support a systematic, geomorphologically-informed approach river management practice. Done properly, this supports a whole-of-government approach. After all, fragmented science can only engender fragmented management.

## Supporting information

**S1 Table. Total stream length of River Styles in the NSW River Styles database.** Note: the raw data in the master database has been processed to produce this summary.
(DOCX)

**S2 Table. Scores for RSGCI by catchment.** Scores in **bold** are below average.
(DOCX)

**S3 Table. Percentages of stream length in each region and catchment that are in different recovery potential and prioritisation classes in the NSW River Styles database.** Note: the raw data in the master database has been processed to produce this summary.
(DOCX)

## Acknowledgments

Release of the NSW River Styles database is the result of around 20 years of collaboration between the NSW DPIE team and Macquarie University. It has involved the expertise of many practitioners and geomorphic experts to further progress the framework and roll it out across the State. We would particularly like to acknowledge the contributions of David Outhet, Nick Cook, Rob Ferguson, Guy Lampert, Ivars Renfelds, Tim Cohen, Rachel Nanson, Carolyn Young, George Schneider, Elisa Zavadil, Tony Broderick, Fiona Marshall, Paul Bennett, Jenny Weingott, Alison Lewis, Bilal Hossain and David Workman.

## Author Contributions

**Conceptualization:** Kirstie Fryirs, Simon Mould, Gary Brierley.

**Data curation:** Kirstie Fryirs, Fergus Hancock, Michael Healey, Simon Mould, Lucy Dobbs, Marcus Riches.

**Formal analysis:** Kirstie Fryirs, Fergus Hancock, Michael Healey, Simon Mould.

**Funding acquisition:** Kirstie Fryirs, Fergus Hancock, Allan Raine, Gary Brierley.

**Investigation:** Kirstie Fryirs, Fergus Hancock, Michael Healey, Simon Mould, Lucy Dobbs, Marcus Riches, Allan Raine, Gary Brierley.

**Methodology:** Kirstie Fryirs, Fergus Hancock, Michael Healey, Simon Mould, Gary Brierley.

**Project administration:** Kirstie Fryirs, Fergus Hancock, Allan Raine, Gary Brierley.

**Resources:** Kirstie Fryirs, Fergus Hancock.

**Supervision:** Kirstie Fryirs, Gary Brierley.

**Validation:** Kirstie Fryirs, Fergus Hancock, Simon Mould, Marcus Riches, Gary Brierley.

**Visualization:** Kirstie Fryirs, Fergus Hancock, Michael Healey, Simon Mould, Lucy Dobbs.

**Writing – original draft:** Kirstie Fryirs, Simon Mould, Gary Brierley.

**Writing – review & editing:** Kirstie Fryirs, Fergus Hancock, Michael Healey, Marcus Riches, Allan Raine, Gary Brierley.

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
