## [Decision Letter · Decision Letter 0]

16 Dec 2020

Things we can do now that we could not do before: Developing and using a cross-scalar, state-wide database to support geomorphologically-informed river management

PONE-D-20-30497

Dear Dr. Fryirs,

We’re pleased to inform you that your manuscript has been judged scientifically suitable for publication and will be formally accepted for publication once it meets all outstanding technical requirements.

Kind regards,

Gabriela Ioana-Toroimac, Ph.D.

Academic Editor

PLOS ONE

**Additional Editor Comments (optional):**

All three reviewers accepted your manuscript for publication. Please read their comments below.

The River Styles Framework and lessons learned are an inspiration for other countries, decision makers, and scientists. Your manuscript presents this topic in a clear and mature manner.

I congratulate you for your work.

**Reviewers' comments:**

Reviewer's Responses to Questions

**Comments to the Author**

1. Is the manuscript technically sound, and do the data support the conclusions?

Reviewer #1: Yes

Reviewer #2: Yes

Reviewer #3: Yes

2. Has the statistical analysis been performed appropriately and rigorously? 

Reviewer #1: Yes

Reviewer #2: N/A

Reviewer #3: Yes

3. Have the authors made all data underlying the findings in their manuscript fully available?

Reviewer #1: Yes

Reviewer #2: Yes

Reviewer #3: Yes

4. Is the manuscript presented in an intelligible fashion and written in standard English?

Reviewer #1: Yes

Reviewer #2: Yes

Reviewer #3: Yes

5. Review Comments to the Author

Reviewer #1: This is an example of an article whose type is difficult to classify clearly. However, it raises a very important issue (from the point of view of contemporary fluvial geomorphology) - the creation, management and interpretation of a large database of rivers within a large country. Recent revitalization projects on various types of rivers require just such a set of data.

I believe the manuscript is of interest to PlosOne readers and should be published. Both the editorial and language side are of a high standard. The manuscript contains as many as 16 figures (plus summlementary material) and I believe that their volume is too large in relation to the text. However, this would be a problem for printed articles rather than PDF articles. The figures explain the authors' intentions in detail and are legible. General sound of the manuscript and the professional presentation of the topic do not raise my doubts.

Reviewer #2: The paper is very clear and exhaustive and achieves perfectly the purpose it aim to.

The explanation is complete and the examples and figures help understand the capacity of the organised amount of information collected.

The authors have been very honest both in highlighting the power and relevance of such informative treasure for river management aware decisions and in stressing the limits of applicability, the need to account for future developments in IT and also resources needed. It is also true that such a db is unique and the first of such type.

I think the paper is definitely mature to be published.

Reviewer #3: The paper entitled “Things we can do now that we could not do before: Developing and using a crossscalar, state-wide database to support geomorphologically-informed river management” aims to present an overview of NSW River Styles statewide database, Australia, and to demonstrate the viability of the River Style Framework as a tool in providing “the basis to contextualize, to plan, to be proactive, to prioritise, to set visions, to set goals and to undertake objective, pragmatic, transparent and evidence-based decision making”

This Open Access product synthesizes approx. 20-year long process of systematic inventory of geomorphic river styles along 216,000 km in length of the stream, including the 3rd and upper stream and totaling 802,000 km2. The database contains references to river types, condition assessment and recovery potential in the survey area, being the largest and most comprehensive of its kind both in Australia and in other parts of the world.

Through the highlighted topics and their references, the authors offer a complete perspective on the whole process behind this database: 1) the theoretical background of the whole exercise (River Style Framework, its position in the international context, compared to other river style frameworks) , ii) collaboration with authorities, stakeholders and managers to create this database, iii) main results and unprecedented opportunities for both systematic river management projections at the local, regional and national basin level, as well as the potential for to help situate national policies in relation to wider international / intercontinental programs. It also provides a valuable overview of the limitations and lessons learned during the development and application of this database.

This paper is the culmination of demonstrating the viability and potential of the River Style Framework to provide the geomorphological foundation for river management. The NSW River Style database represents the implementation of the entire theoretical scaffold developed by the authors, the result being an inventory of the state of the rivers at a scientific visionary level. On the same level of value is the demonstration of how the authors initiated and developed the collaboration with the authorities, for the official takeover of the River Style Framework in river mapping.

The material has the potential to become a reference work in the field of river geomorphology, both by the real value of database analysis and by the elegant demonstration of the applicability of the River Style Framework. I appreciate the material as excellent and recommend publishing it as it is.

Two observations that can contribute to the improvement of the material:

- Fig 10 has a relatively low resolution

- References: 30 and 31 are duplicates

6. PLOS authors have the option to publish the peer review history of their article (what does this mean?). If published, this will include your full peer review and any attached files.

Reviewer #1: No

Reviewer #2: No

Reviewer #3: No

---

## [Editor Report · Acceptance letter]

28 Dec 2020

PONE-D-20-30497 

Things we can do now that we could not do before: Developing and using a cross-scalar, state-wide database to support geomorphologically-informed river management 

Dear Dr. Fryirs:

I'm pleased to inform you that your manuscript has been deemed suitable for publication in PLOS ONE. Congratulations! Your manuscript is now with our production department. 

Kind regards, 

on behalf of

Dr. Gabriela Ioana-Toroimac 

Academic Editor

PLOS ONE